# Melting of recycled ancient crust responsible for the Gutenberg discontinuity

Jia Liu [1]*, Naoto Hirano [2], Shiki Machida[3], Qunke Xia [4], Chunhui Tao[1,5], Shili Liao[1], Jin Liang[1], Wei Li[1], Weifang Yang[1], Guoying Zhang[1] & Teng Ding[1,6]

A discontinuity in the seismic velocity associated with the lithosphere-asthenosphere interface, known as the Gutenberg discontinuity, is enigmatic in its origin. While partial mantle melts are frequently suggested to explain this discontinuity, it is not well known which factors critically regulate the melt production. Here, we report geochemical evidence showing that the melt fractions in the lithosphere-asthenosphere boundary were enhanced not only by accumulation of compacted carbonated melts related to recycled ancient marine sediments, but also by partial melting of a pyroxene-rich mantle domain related to the recycled oceanic eclogite/pyroxenites. This conclusion is derived from the first set of Mg isotope data for a suite of young petit-spot basalts erupted on the northwest Pacific plate, where a clearly defined Gutenberg discontinuity exists. Our results reveal a specific linkage between the Gutenberg discontinuity beneath the normal oceanic regions and the recycling of ancient subducted crust and carbonate through the deep Earth.

[1] Key Laboratory of Submarine Geosciences, Second Institute of Oceanography, Ministry of Natural Resources, 310012 Hangzhou, China. [2] Center for Northeast Asian Studies, Tohoku University, 41 Kawauchi, Aoba-ku, Sendai 980-8576, Japan. [3] Chiba Institute of Technology, Ocean Resources Research Center for Next Generation, Chiba 275-0016, Japan. [4] School of Earth Sciences, Zhejiang University, 310027 Hangzhou, China. [5] School of Oceanography, Shanghai Jiao Tong University, Shanghai 200240, China. [6] School of Oceanography, Hohai University, Nanjing, China. *email: liujia@sio.org.cn

The lithosphere–asthenosphere boundary (LAB) beneath ocean basins separating the high-velocity lithosphere of varying thickness and the underlying low velocity zone (LVZ) is of utmost importance in understanding the geochemical and geodynamic evolution of our planet[1]. The large decrease in the seismic wave velocity up to 6–9%, usually at the top of the LVZ beneath the oceanic lithosphere, is referred to as the Gutenberg (G) discontinuity[2,3]. In some oceanic regions, this discontinuity coincides with the expected LAB depth[3,4]. Although partial melting of the mantle is frequently suggested as an explanation for the G discontinuity[4–8], critics[9,10] argue that the partial melts of the mantle away from oceanic ridges are too low in volume to produce sufficient effects on the seismic velocity. Based on the parameterizations of experiments constraining the influence of $CO_2$ and $H_2O$ on silicate melting, and the experimental stability of carbonatite in the upper mantle, Hirschman[11] calculated the stability of partial melts in the LVZ beneath the oceanic lithosphere. The results show that for the mantle sources of mid-ocean ridge basalt (100 ppm $H_2O$ and 60 ppm $CO_2$) with mantle potential temperature of ~1350 °C, the maxima melt fraction would be <0.1% beneath older (>40 Ma) lithosphere. This melt fraction is considerably lower than that required to explain the occurrence of the G discontinuity[6]. This means that the melt

fraction should be efficiently enhanced by some ways, for instance the accumulation of deep-mantle-derived carbonatite/carbonated silicate melts by deformation or compaction of the mantle[12,13], or the melting anomaly associated with an extra-volatile flux or the presence of a pyroxene-rich domain[11], if the partial melting scenario is valid. This reasoning correlates the formation of partial melts responsible for the G discontinuity with the Earth's deep carbon cycle and lithological heterogeneities in the mantle.

The northwest Pacific is one of the typical regions where several high-quality seismic studies of the LAB of the subducting Pacific plate have been performed[6,7,14]. Kawakatsu et al.[6] obtained a clear receiver function image of a sharp surface showing S-wave velocity reduction (up to 7–8%) and explained it as the presence of the G discontinuity at a depth of ~80 km beneath the seafloor (Fig. 1). This sharp drop in the shear wave velocity was supposed to be caused by horizontal melt-rich layers embedded in the meltless mantle, with an average melt fraction of 1.25–0.25%[6]. The presence of petit-spot volcanoes laying on the 131–136 Ma subducting NW Pacific plate in front of the Japan trench (sites A, B, C in Fig. 1) have been reported by refs. [15,16]. Recent geochronological and geochemical data together with results of experimental petrology show that these volcanoes represent partial melts from the LAB[16–18]. Thus, these petit-spot

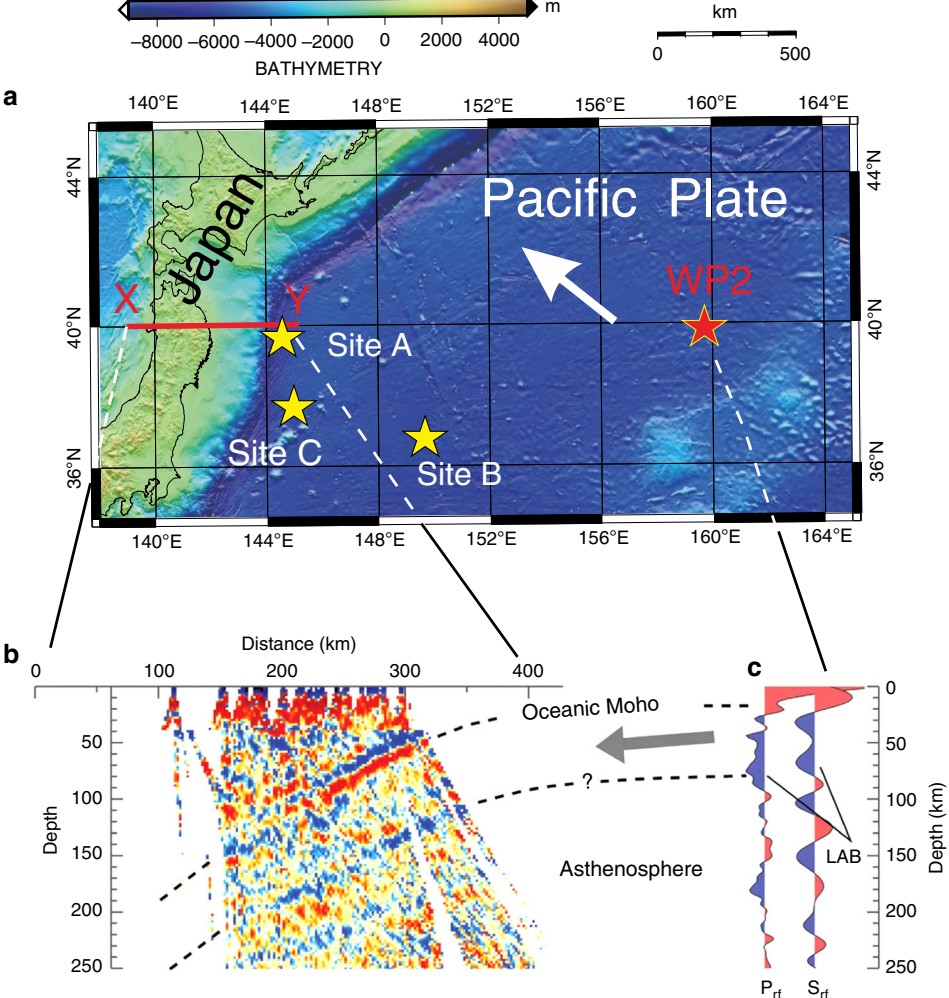

**Fig. 1 Bathymetric maps for NW Pacific showing the position of the petit-spot volcanoes investigated in this study and the RF image of the subducting Pacific plate beneath northeast Japan. a** Sites A, B, and C showing the location of petit-spot basalts. WP2 is a seafloor borehole station and the white arrow indicates the moving direction of the Northwestern Pacific plate. **b** P-RF image along the profile X–Y using dense land seismic data of Hi-net from Japan, adapted from ref. [6] Reprinted with permission from AAAS. The two dashed lines show the oceanic Moho and the oceanic LAB, respectively. The red and blue colors indicate a velocity increase (from shallow to deep) and decrease at the point, respectively.

lavas provide a unique opportunity to integrate both geophysical and geochemical observations in attempts to investigate the origin of melt layers beneath an old oceanic LAB. The previous geochemical studies have shown that these petit-spot basalts have Sr-Nb-Pb isotopes and trace element characteristics similar to those of the EM1-type (enriched mantle) oceanic island basalts, which may indicate the presence of small-scale heterogeneities in the upper mantle caused by small blobs of recycled ancient plate materials[16,19]. The geochemical constraints on the origin of petit-spot volcanoes has also indicated the importance of $CO_2$-fluid or carbonate in partial melting of the asthenosphere[17,18]. However, the linkage between the small-scale mantle lithological heterogeneities and the global carbon recycling, and their associations with partial melting in the LAB remains unclear.

Given that Mg is a major component in all mantle-derived magmas and its stable isotope composition can be fractionated by surficial and low-temperature processes but not by high-temperature magmatic processes like partial melting and fractionation[20–23], the Mg isotopes may provide new insights into the origin of the enriched blobs and $CO_2$ in the asthenosphere. Here, we report trace element abundances and Sr–Nd–Pb compositions and $\delta^{26}Mg$ values for 15 basalt samples from the petit spot volcanoes occurring at Sites A, B, and C, and try to decipher the relationships of carbon recycling, mantle heterogeneities and the genesis of partial melts in the mantle responsible for the G discontinuity. Our data shows that the partial melts enhancement responsible for the generation of G discontinuity are caused by accumulation of compacted carbonated melts related to recycled ancient marine sediments, and also by the partial melting of a pyroxene-rich mantle domain related to the recycled oceanic eclogite/pyroxenites.

## Results

**Basic geochemistry of the petit-spot basalts**. The studied petit-spot basalt samples are vesicular and very fresh. Their petrological and geochemical characteristics have previously been described in refs. [16,19,24]. Major element and newly measured trace element abundances, Sr–Nd–Pb isotopes compositions and whole-rock $\delta^{26}Mg$ values (relative to DSM-3) of the samples are reported in Supplementary Data 1–4. The $SiO_2$ content of the basalts are from 39.4 to 49.2 wt.%. Their total alkali ($Na_2O + K_2O$) contents range from 2.72 to 7.7 wt.% and $K_2O/Na_2O$ ratios are generally higher than 1.0 (Supplementary Fig. 1a; Supplementary Data. 1). The basalts are relatively primitive, with their bulk-rock MgO contents ranging from 8.44 to 13.17 wt.%, except for one sample (6K#879) with a lower MgO content of 6.6 wt.%. In general, the basalts show OIB-like trace element patterns and EM1-like Sr–Nd–Pb isotope compositions (Supplementary Fig. 1b, c). The $^{206}Pb/^{204}Pb$ ratios vary from 16.941 to 17.924 (Fig. 2), being considerably lower than those in the Pitcairn Island basalts representing the EM1 end-member in the whole OIB database[25]. The lead isotopic compositions show linear correlations with both major and trace element ratios, such as $K_2O/TiO_2$, (Sm/Yb)n (n indicates primitive mantle normalization), Ce/Pb, Ti/Eu, and Nb/Ta (Fig. 2).

**Mg isotopic compositions**. In the plot of $\delta^{25}Mg$ vs. $\delta^{26}Mg$ (Supplementary Fig. 2), the basalts and the USGS standards fall along the terrestrial equilibrium mass fractionation line with a slope of 0.521[26]. The nine samples from Site A show $\delta^{26}Mg$ values from $-0.22 \pm 0.05$ to $-0.45 \pm 0.04$ ‰ (2SD), with more than half of them being lower than the range defined by peridotitic mantle and the global mid-ocean ridge basalts (MORBs) and ocean island basalts (OIBs) database ($-0.25 \pm 0.07$‰, 2SD[21]) (Fig. 3a). The $\delta^{26}Mg$ values of the two samples from Site B are

$-0.35 \pm 0.02$‰ (2SD) and $-0.33 \pm 0.06$‰ (2SD), which are also slightly lower than that of the mantle based on oceanic basalts and peridotites (Fig. 3a). For the low-$SiO_2$ samples from Site C, $\delta^{26}Mg$ varies from $-0.30 \pm 0.05$ to $-0.45 \pm 0.04$‰ (2SD). The lower limit of these $\delta^{26}Mg$ values is also lower than that of the Pitcairn Island EM1 basalt and slightly higher than that of the continental EM1 basalts in northeastern China (Fig. 3a). In addition, the $\delta^{26}Mg$ values exhibit positive correlations with $TiO_2$, with the exception of one sample from Site C (#1392R10), and $^{206}Pb/^{204}Pb$, Ce/Pb, U/Pb, and (Sm/Yb)n ratios (Figs. 3b and 4).

## Discussion

There are multiple factors that could lead to variation in the Mg isotope composition of mantle-derived basalts. In our case, the role of seawater alteration in the formation of subnormal $\delta^{26}Mg$ can be excluded, because there is no clear correlation between $\delta^{26}Mg$ and loss on ignition (LOI), and the lowest $\delta^{26}Mg$ is preserved in the sample with least alteration (see Supplementary Fig. 3). In addition, the bulk rock Ba/Rb and Th/U ratios are positively correlated with LOI (Supplementary Fig. 3), which argue against the significant role of seawater alteration in resetting the composition of our samples (if seawater alteration does matter, negative correlations should be expected, because the Rb and U would be added in the rock in such process[27]). Fractional crystallization of mafic minerals, such as olivine or clinopyroxene, would not be the main reason for the observed low and heterogeneous $\delta^{26}Mg$ values because, on one hand, the Mg isotope fractionation between these minerals and basaltic melt at magmatic temperatures is rather limited ($\pm 0.07$‰[28]) and, on the other hand, all the samples except one (#879-R3A, MgO = 6.6wt.%) are relatively primitive (MgO from 8.4 to 13.2 wt.%). The marine sediments rich in dolomite in the seabed on which the petit-spot volcanoes erupted could have abnormally low $\delta^{26}Mg$ values[20] and therefore, assimilation of these sediments during the extrusion of the basalts could be responsible for their light Mg isotope compositions. These low $\delta^{26}Mg$ sediments usually show rather radiogenic $^{206}Pb/^{204}Pb$ and low Ce/Pb ratios[20,29]. As shown in Fig. 2a, the spreading of Pb isotopes of our samples argue against significant role as the contaminators. In addition, the lack of positive correlation between Ce/Pb and $\delta^{26}Mg$ is inconsistent with this possibility (Fig. 4b) also does not support. As shown in Fig. 3a, the $\delta^{26}Mg$ values of the basalts do not correlate with their Mg#, which argues against the role of assimilation of carbonated oceanic crust in the formation of the subnormal $\delta^{26}Mg$ values.

During partial melting of a garnet-bearing peridotite or pyroxenite source, the largest fractionation of $\delta^{26}Mg$ between the partial melt and solid residue can be expected to be <+0.1‰[30]. In addition, the $\delta^{26}Mg$ values in our samples show positive correlations with the Pb isotope ratios (although the correlation coefficient (R) is relatively low, the t-test shows that the correlation is significant in the 95% confidence range) and the elemental ratios that are not sensitive to the degree of partial melting (e.g., Ce/Pb; Fig. 4b). Thus, a simple partial melting process cannot explain the $\delta^{26}Mg$ values that are as low as $-0.45 \pm 0.06$‰. In addition, the linear correlation between $\delta^{26}Mg$ and Ce/Pb argues against the role of interaction of basalt with lithospheric mantle, because the interaction with lithospheric mantle would largely buffer the Mg isotope composition but would not change the Ce/Pb ratios significantly. A mantle source rich in ilmenite could produce subnormal $\delta^{26}Mg$[31], which seems to be compatible with the negative correlation between $\delta^{26}Mg$ and $TiO_2$ in our samples (Fig. 3b). However, several lines of evidence argue against this scenario. First, the $TiO_2$ content of our samples are significantly lower than that of the low-$\delta^{26}Mg$ high-Ti lunar

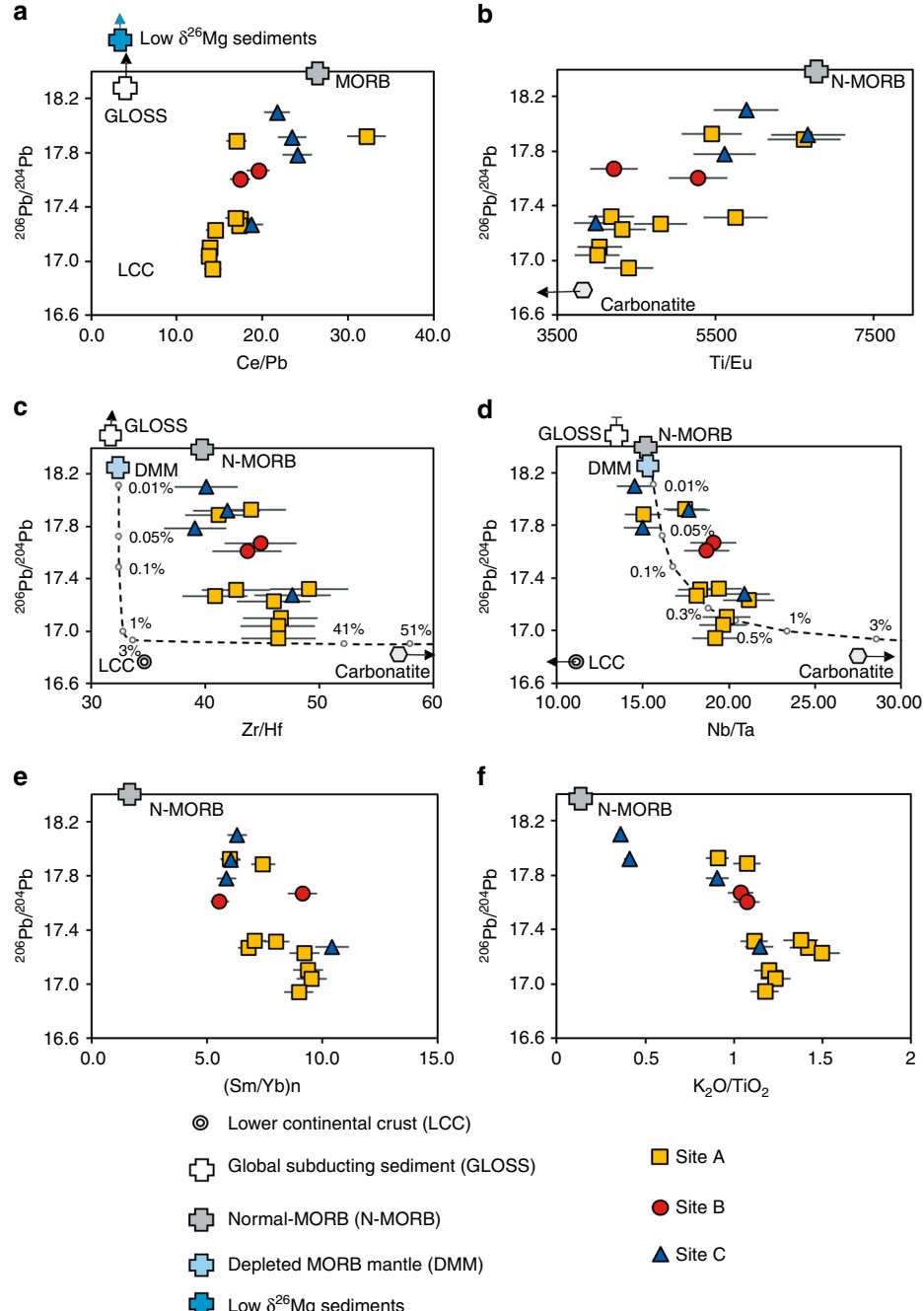

**Fig. 2 Diagrams showing correlations between $^{206}Pb/^{204}Pb$ and trace and major elemental ratios for petit-spot basalts from the NW Pacific plate.**
**a** the comparison of measured $^{206}Pb/^{204}Pb$ with the Ce/Pb ratio for the basalts from Site A, B, and C. The 2SD of the $^{206}Pb/^{204}Pb$ are from 0.002 to 0.02, which are all smaller than the scale of the symbols. The relative uncertainty (2SD) of the Ce/Pb ratio is 7%, which is calculated as $\sqrt{\sigma_{Ce}^2 + \sigma_{Pb}^2}$ ($\sigma_{Ce}$ and $\sigma_{Pb}$ are the relative uncertainty for Ce and Pb measurement, both of which is around 5%). The corresponding value of lower continental crust (LCC), global subducting sediments (GLOSS), the low $\delta^{26}Mg$ marine sediments in the subducting oceanic slab, the normal-MORB (N-MORB), the depleted MORB mantle (DMM), and the carbonatite melt are shown for comparisons (data are from[20,58–61]). **b** the comparison of measured $^{206}Pb/^{204}Pb$ with Ti/Eu ratio. The relative uncertainty (2SD) of the Ti/Eu ratio is 7%. The white hexagon and the dark real arrow shows the composition of the carbonatite, and the data are from ref. [60]. **c** the comparison of measured $^{206}Pb/^{204}Pb$ with Zr/Hf ratio. The relative uncertainty (2SD) of the Zr/Hf ratio is 7%. The dashed curve shows the simulated mixing between the DMM and the carbonatite melt (data for these two endmembers are from ref. [59,60], and the numbers are the percentage by weight of the carbonatite melt. **d** the comparison of measured $^{206}Pb/^{204}Pb$ with Nb/Ta ratio. The relative uncertainty (2SD) of the Nb/Ta ratio is 7%.The dashed curve shows the simulated mixing between the DMM and the carbonatite melt (data for these two endmembers are from ref. [59,60], and the numbers are the percentage by weight of the carbonatite melt. **e** the comparison of $^{206}Pb/^{204}Pb$ with primitive mantle normalized Sm/Yb ratio. The relative uncertainty (2SD) of the Sm/Yb ratio is 7%. The primitive mantle composition is from ref. [62]. **f** the comparison of measured $^{206}Pb/^{204}Pb$ with $K_2O/TiO_2$ ratio. The relative uncertainty (2SD) of the $K_2O/TiO_2$ ratio is 7%.

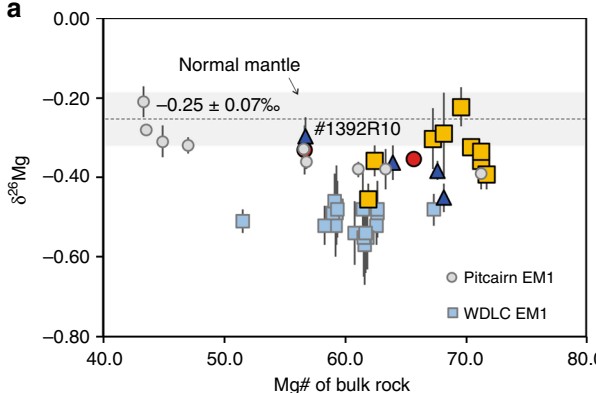

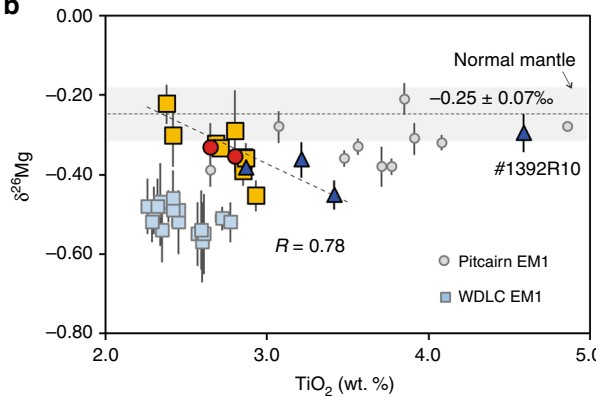

**Fig. 3** $\delta^{26}Mg$ **vs. MgO and TiO$_2$ for petit-spot basalt samples. a** The comparison of $\delta^{26}Mg$ with bulk rock Mg#. The gray box with a dashed line shows the Mg isotopic composition of the global MORB and OIBs (ref. [21]), referred to as the Normal Mantle value here. The Mg isotope composition of the Pitcairn Island EM1 basalts and the continental intraplate EM1 basalts from northeastern China (Wudalianchi, WDLC) are shown for comparison (ref. [39,41]). All error bars represent 2 standard deviations. Symbols for petit-spot basalts are the same as in Fig. 2. #1392R10 marks the sample from Site C, which is out of the trend between $\delta^{26}Mg$ and TiO$_2$. **b** The comparison of $\delta^{26}Mg$ with the TiO$_2$ content of the bulk rocks. All the symbols are the same to a and Fig. 2. R is the correlation coefficient of the linear regression for all the petit-spot basalt samples except #1392R10, which clearly stands outside the trend.

basalts (2.72 ± 0.17 wt.% vs. >6 wt.%[31]). Second, because the partition coefficients of Nb and Zr are lower than those of Ta and Hf for ilmenite in mafic magmas[32], one would expect negative correlation of TiO$_2$ between Nb/Ta and Zr/Hf if the mantle source contained a lot of ilmenite, but these are not observed. Overall, we can conclude that the subnormal $\delta^{26}Mg$ values in our samples are not due to partial melting and other processes during magma transport, but rather they are results of a Mg isotopic heterogeneity in the shallowest asthenosphere, which could be associated with the presence of recycled material.

As shown in Fig. 2, the Pb isotope compositions are linearly correlated with certain major and trace element ratios in the petit-spot basalts from different sites. These correlations cannot not be explained by assimilation of seabed sediments during the magma transport and/or contamination of the DMM mantle source by components with low Pb isotope ratios, such as ancient lower continental crust (LCC) or carbonatite (Fig. 2a, b, d). Instead, they could be explained by mixing between two batches of magma with distinct chemical characteristics. The low Ce/Pb ratios for the end-member with low $^{206}Pb/^{204}Pb$ seems (Fig. 2a) to be consistent with the components suggested by the classical

models for the genesis of the EM1 signature[33–36], namely recycled ancient crustal and/or sedimentary material or a segment of ancient metasomatized lithospheric mantle. However, these models would not be responsible for the EM1 signature of the petit-spot basalts, as several lines of evidence indicate that this end-member was most likely derived from a mantle source metasomatized by carbonatite melts originated from ancient carbonate-bearing marine sediments. First, the low-$^{206}Pb/^{204}Pb$ end-member shows low Ti/Eu ratios coupled with high Nb/Ta and Zr/Hf ratios (Fig. 2b–d), which are not consistent with the compositions of the ancient lower continental crust or typical marine sediments (Fig. 2). Instead, they fit more with the composition of carbonatites or carbonated silicates (Fig. 2c–d). Second, the CO$_2$ content of the primary magma of the studied petit-spot lavas (inferred from the measurement of CO$_2$ and H$_2$O content in the quench glass and the modeling of degassing process based on the CO$_2$–H$_2$O solubility model) are higher than 10 wt.%[17,37], which indicates that the primary magmas were highly carbonated. Third, the low-$^{206}Pb/^{204}Pb$ end-member shows both high K$_2$O/TiO$_2$ and (Sm/Yb)n ratios (Fig. 2e, f) and could not be formed by partial melting of a common or carbonated peridotite and eclogite source. Instead, the Pb isotope composition is more compatible with partial melting of carbonated pelite at a pressure higher than 8 GPa[38]. In the following discussion, this sediment is referred to as the Carbonated Marine Sediment (CMS) end-member.

While the correlations of Pb isotopes and incompatible major and trace element ratios (Fig. 2a) might suggest simple mixing between N-MORB-like melts and carbonated silicate melts, the correlations between $\delta^{26}Mg$ and $^{206}/Pb^{204}Pb$ and Ce/Pb ratios indicate a different scenario. The high-$^{206}/Pb^{204}Pb$ end-member shows a clearly subnormal $\delta^{26}Mg$ value (Fig. 4a), which argues against simple derivation from a DMM source with $\delta^{26}Mg$ of ~−0.25 ± 0.07‰. In the previous discussion, we suggest that this subnormal Mg isotope composition could only be attributed to a heterogeneous mantle source. There are several potential candidates to cause these heterogeneities, namely subducted sedimentary carbonates[20], recycled oceanic eclogites that had experienced carbonation in their early sub-duction stage but were decarbonated during their later evolu-tion[39], or the recycled eclogites experienced Mg-Fe exchange with surrounding peridotites[40]. These candidates have been suggested to explain the $\delta^{26}Mg$ anomalies in the continental Wudalianchi EM1 basalts in northeast China[41] and the Pitcairn Island EM1 and Cenozoic intraplate HIMU basalts in New Zealand[39,42]. The N-MORB-like trace element ratios, such as Zr/Hf, Ti/Eu, and Nb/Ta, in this high-$^{206}/Pb^{204}Pb$ end-member (Fig. 2b–d) are not consistent with the origin from a sedi-mentary carbonate source, but fit more with the eclogitic component. This is also supported by the observation that the subnormal $\delta^{26}Mg$ component is associated with high Ce/Pb and U/Pb ratios and low (Sm/Yb)n ratios (Fig. 4b–d), which are characteristic of subducted eclogites/pyroxenite[43]. In the fol-lowing discussion, this component is referred to as the Eclogite/pyroxenite end-member. Overall, according to the discussion above, our geochemical data indicate that the mantle source of the petit-spot basalts contain recycled oceanic plate materials with different origin.

As mentioned in the beginning, an increase in the melt fraction beneath the LAB could be reached through accumulation of carbonatite melts migrated from greater depths replenished by the melts filtered out from a melt-bearing LVZ, or the relatively high melt fraction derived from enriched mantle either with enriched volatile concentrations or pyroxene-rich lithologies[11]. The identification of the recycled components in the mantle sources of the petit-spot basalts would give us information on the

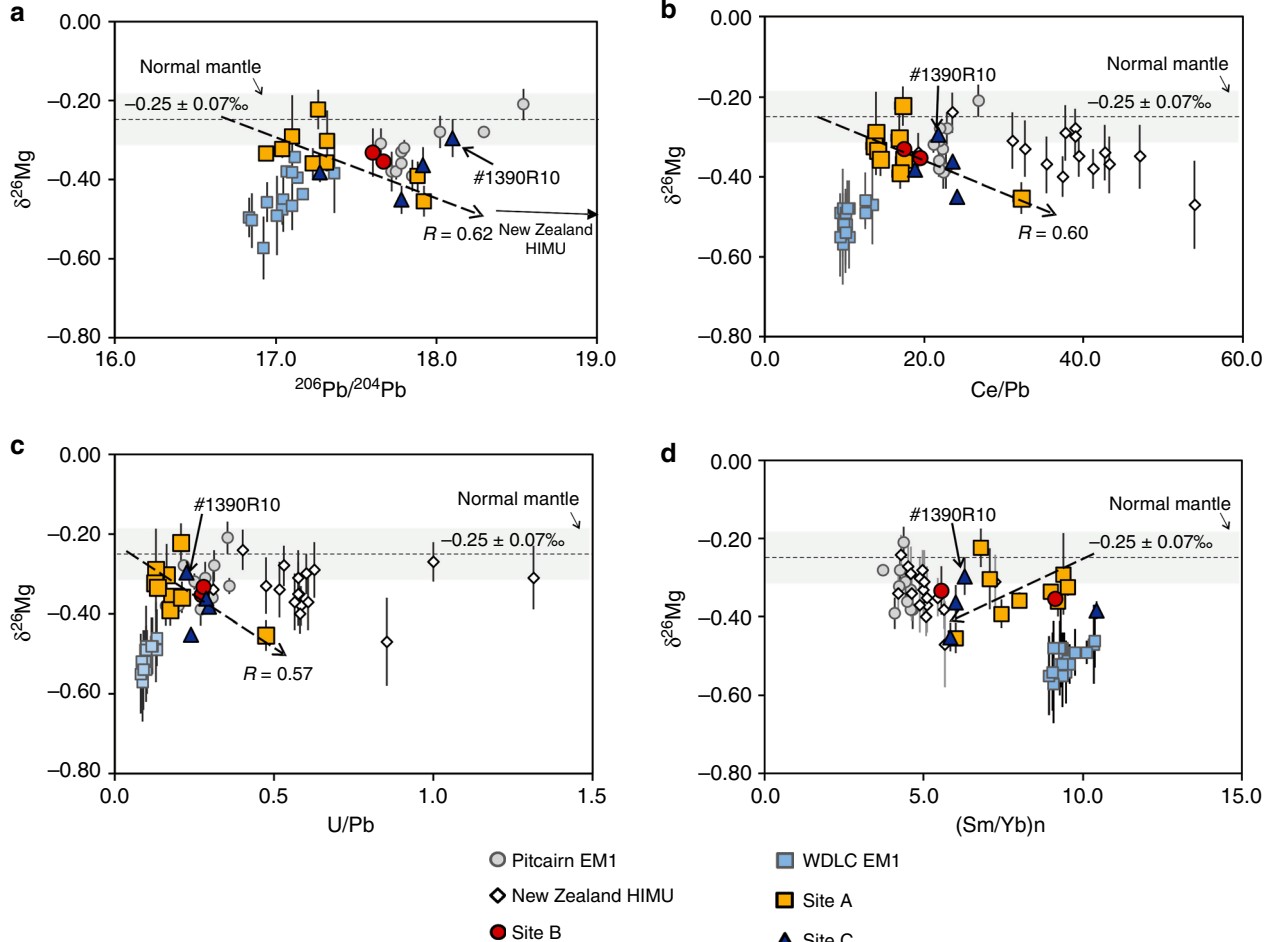

**Fig. 4 δ²⁶Mg vs. elemental and radiogenic isotopic ratios for petit-spot basalts. a** the comparison of δ²⁶Mg with ²⁰⁶Pb/²⁰⁴Pb ratio. The symbols have been explained below the figure, and they are the same as in Fig. 3. The dark arrow points to the New Zealand HIMU samples are from the South Island and Antipodes Islands, for which the ancient recycled eclogites have been suggested to explain the origin low δ²⁶Mg isotope values coupled with high Pb isotope ratios[42]. The sample from Site C, labeled as #1390R10, is marked because it plots outside the TiO₂-δ²⁶Mg linear trend shown in Fig. 3b. The dashed line with an arrow indicates the linear regression line (regression coefficients shown) fitting with all petit-spot lavas except #1390R10. **b** the comparison of δ²⁶Mg with Ce/Pb ratio. The white diamonds are the New Zealand HIMU samples. Others are the same from **a**. **c** the comparison of δ²⁶Mg with U/Pb ratio. **d** the comparison of δ²⁶Mg with primitive mantle normalized Sm/Yb ratio.

mechanism of melt layer formation and enhance our understanding of the origin of the G discontinuity. First of all, the EM1-like Pb isotope signatures and subnormal Mg isotope values of the petit-spot basalts (Figs. 2, 3) indicate that the melt layer is not solely generated by accumulation of partial melts of a normal MORB-like mantle source caused by a sharp decrease in the water solubility with depth[44], breakdown of pargasite[45], or the presence of trace amounts of CO₂[46]. Instead, these isotope characteristics fit more with a mantle source associated with enriched CO₂ and/or lithological heterogeneities.

In the previous discussion, we conclude that CO₂ enrichment (Carbonated end-member, caused by carbonated silicates or carbonatite melts with high Zr/Hf, Nb/Ta, and low Ti/Eu ratios) was closely associated with a component with low (EM1-like) ²⁰⁶Pb/²⁰⁴Pb ratios. It has been well known that the EM1 signature needs a low-U/Pb reservoir isolated for a long time (>1Ga)[19], which is much longer than the age of the present Pacific plate (~>130 Ma). A possible way to store this carbonated component in the oceanic mantle far from a hotspot and oceanic ridge is in the form of reduced carbon, such as diamond or metal-carbide, in the very reduced lower part of the upper mantle[47], until its recent

partial melting due to regional mantle upwelling[4]. In this scenario, the carbonanites or carbonated silicates would form a low-degree porous melts flowing in the peridotite matrix[39,48,49] (Fig. 5) and accumulating in the top part of the asthenosphere and thus contributing to the formation of the G discontinuity[46,50]. With a diffusion rate of Mg of the order of $10^{-16}$ m²/s in the major silicates (olivine, opx, garnet)[51], the diffusion distance would be 0.02 m in the time scale of 0.1 Ma (see Methods). This porous flow could thus easily lead to complete Mg isotope buffering by the peridotite matrix (see Methods) and induce decoupling of Mg isotopes from strong carbonate-indicating element ratios (high Zr/Hf, Nb/Ta, and low Ti/Eu ratios) for the CMS end-member (Figs. 2 and 4), which is different from their EM1 counterparts in northeastern China (WDLC EM1) and Pitcairn hotspot oceanic island basalts. It should be noted that this Mg isotopic buffering effect is in large contrast to the large Mg isotopic fractionation (over 1‰) in the porous flow reaction front of an exhumed contact between rocks of subducted crust and serpentinite in the Syros melange zone[23]. The main reason for this large Mg isotopic fractionation is that only the grain-boundary diffusion was considered, because the

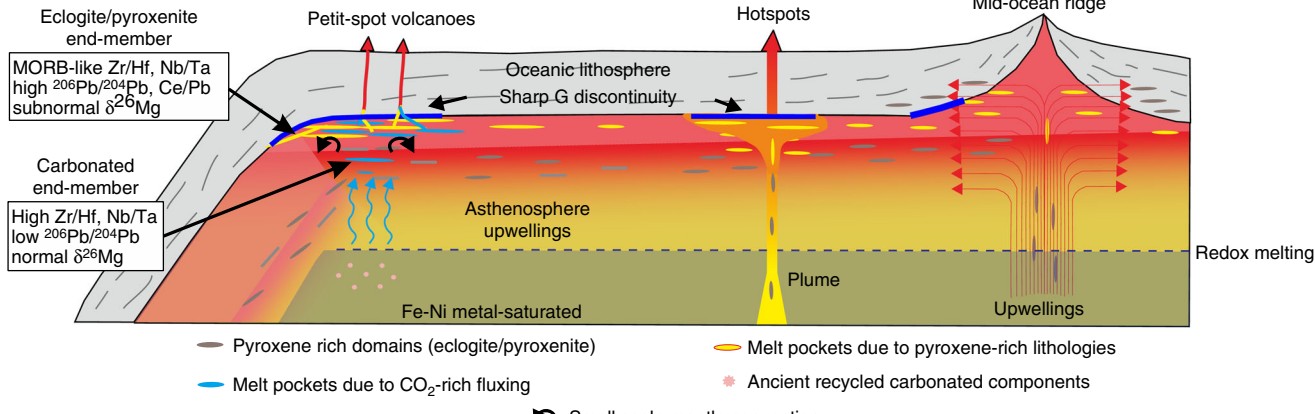

**Fig. 5 Schematic cross-section of the oceanic mantle illustrating potential mechanisms that could concentrate partial melts below the oceanic LAB.**
The G discontinuity (bold dark blue lines) corresponds to the melt layer enhanced by the stagnation of partial melt at the base of the lithosphere. The dark gray lenses show pyroxene-rich heterogeneities due to recycling of ancient crustal materials. They will develop melt pockets in the shallow LVZ (yellow lenses) due to different scale of upwelling of the asthenosphere at ridges, hotspots or near the subduction zone[4], and stretched horizontally due to the mantle flow[11]. Beneath the old oceanic lithosphere where petit-spot basalts erupted, two end-members of recycled components (recycled carbonated marine sediments and the recycled oceanic eclogite/pyroxenite) contribute to the formation of partial melts that finally generate melt layers beneath the LAB and cause the origin of the G discontinuity. See the text for the details of the model.

Mg transportation re-equilibrium between the parous fluids and the matrix minerals was sluggish due to the slow Mg isotopic diffusion in the solid in low temperature (~450 °C)[23].

As discussed above, the Eclogite/pyroxenite end-member (higher Ce/Pb and U/Pb and lower (Sm/Yb)n) could be attributed to recycled oceanic crust, which acquired its anomalous Mg isotope composition before subduction. The preservation of the Mg isotopic anomaly in this end-member (Fig. 4a) argues against the same porous flow as in the case of the CMS end-member. Instead, the survival of the subnormal Mg isotope composition requires that the melt efficiently resisted diffusion or reaction with the ambient mantle. A convenient solution for this requirement is that low $\delta^{26}$Mg remained unchanged in solid blobs in the matrix peridotites or melt layers were blocked by some low-permeability interfaces. Our diffusion modeling (see Method) shows that in the solid state, the $\delta^{26}$Mg anomaly of a recycled pyroxenite/eclogite of ~10 m could be preserved for more than 1 billion years. This time period is long enough for the subduction of ancient oceanic lithosphere and the subsequent formation of enriched domains (as pyroxenite) stretched into high-aspect-ratio lenses by mantle convection[52]. In addition, due to the large melt/pyroxene dihedral angles and the strong tendency for garnet to form facets, the resulting low permeability of the eclogite/pyroxenites would retain the partial melt and prevent it from interacting with the ambient mantle[11]. Thus, the observed negative Mg isotope anomaly supports the presence of melt lenses beneath an old lithosphere generated by partial melting of pyroxene-rich lithologies in the peridotite-dominant asthenosphere[11]. This pyroxene-rich domain represents elongated strips of subducted oceanic lithosphere formed by stretching and thinning by the normal and shear strains in the convecting mantle (so called "marble cake mantle"[52]), or the cooling down of the partial melts produced by the upwelling mantle beneath the mid-oceanic ridges[11] (Fig. 5).

In summary, we show that recycling of crustal materials into the asthenosphere not only transport the melting-favoring volatiles, but also contribute to the pyroxene-rich lithology of the upper mantle. These components contribute to the mantle carbon release, radiogenic isotopic heterogeneities (e.g., EM1 signature), partial melt enhancement beneath the LAB, and consequently the generation of the oceanic G discontinuity (Fig. 5). This scenario would also be applicable to the G discontinuities beneath other

old oceanic lithosphere LAB, given that the petit-spot lavas with enriched geochemical features are also found on the subducting Pacific plate near the Chile trench[53].

## Methods

**Mg isotope analyses.** Isotopic ratios of Mg were measured at the University of Science and Technology of China (USTC), following the procedure described in ref. [54]. Repeated analyses of whole-rock powders showed an excellent consistency of Mg isotope data. Approximately 1 ml aliquots of the stock bulk solution were used for chromatographic separation. Magnesium was separated in pre-cleaned Bio-Rad cation resin (AG50W-X12) columns using 2 N $HNO_3$ + 0.5 N HF and 1 N $HNO_3$. Solutions before and after the Mg cut were collected to check the recovery of Mg by comparing the amount of Mg in the different cuts. A pure Mg solution with a high yield was obtained by two passes through the column. Isotope ratios were measured via multicollector inductively coupled plasma mass spectrometry (MC-ICP-MS) using a Thermo Scientific Neptune Plus instrument. A "wet" plasma, using a quartz dual cyclonic-spray chamber and an ESI 50 μL min−1 PFA MicroFlow Teflon nebulizer (Elemental Scientific Inc., U.S.A.) was utilized in the mass spectrometers. The international whole-rock standards (BCR-2, BHVO-2) were analyzed for trace the accuracy. The in-house Mg solution standard (IGGMg1, IEE) were repeatedly analyzed during the session. These data show an excellent agreement with the recommended literature values, highlighting the reliability of the Mg isotope data. The 2SD of the reported $\delta^{26}$Mg values are mostly lower than 0.05‰.

**Trace element analysis of bulk-rock samples.** Trace element analyses were performed using an inductively coupled plasma mass spectrometry (Agilent Technologies 7700x quadrupole ICP-MS) at Nanjing FocuMS Technology Co. Ltd. Powdered rock samples with a weight of 0.0400 g were digested completely with 0.5 mL of $HNO_3$ and 1.0 mL of HF in a tightly sealed, 7-mL Teflon PFA screw-cap beaker (Savillex®) heated for 48 h on a hot plate at 195 °C and then evaporated to dryness for more than 5 h at 110 °C. The evaporates were diluted to a mass ratio of 1:2000 and then nebulized into the mass spectrometer. To monitor the data quality, the same procedures were followed for the used USGS basaltic geochemical reference materials (BIR-1, BCR-2, BHVO-2). Measured values of these reference materials were compared with preferred values in the GeoReM database (Jochum and Nohl,2008; http://georem.mpch-mainz.gwdg.de). The deviations were better than ±10% and ±5% for the elements exceeding 10 and 50 ppm in abundance, respectively.

**Sr–Nd–Pb isotopic analyses.** The same rock powders that were used to determine major and trace element compositions were also used for bulk-rock Sr, Nd, and Pb isotopic analysis. By following the procedures described by ref. [55], Sr–Nd–Pb isotope analyses were performed using multicollector inductively coupled plasma mass spectrometry (MC-ICP-MS; Nu Plasma II) at Nanjing FocuMS Technology Co. Ltd. Approximately 20–110 mg of powdered rock or glass samples were digested completely using 1 mL of $HNO_3$, 2.5 Ml of $HClO_4$, and 2.5 mL of HF in tightly sealed 7-mL Teflon PFA screw-cap beakers. They were heated for 18 h on a

hot plate at 180 °C, then evaporated for more than 5 h at 110 °C, and finally heated at 180 °C to make the residue almost completely dry. Element separation was accomplished from the same digestion solutions by two-step column chemistry. Strontium, REEs, and Pb were extracted with the BioRad AG50W × 8 combined with Sr Spec resin. Then, Nd was further isolated from the REEs using Ln resin (Eichrom Technologies Inc.).

The Sr-, Nd-, Pb-bearing elutes were dried down and re-dissolved in 1.0 ml of 2-wt% $HNO_3$. Small aliquots of each solution were analyzed using an Agilent Technologies 7700x quadrupole ICP-MS instrument to determine the exact contents of Sr, Nd, and Pb. Diluted solution (50 ppb Sr, 50 ppb Nd, 40 ppb Pb doping with 10 ppb Tl) were introduced into the MC-ICP-MS by the Teledyne Cetac Technologies Aridus II desolvating nebulizer system (Omaha, Nebraska, USA). The measured isotope ratios were corrected for instrumental fractionation by applying the following measured isotope ratios: $^{86}Sr/^{88}Sr$ 0.1194, $^{146}Nd/^{144}Nd$ 0.7219, and $^{205}Tl/^{203}Tl$ 2.3885 (for Pb isotopes). International isotopic standards (NIST SRM 987 for Sr, JNdi-1 for Nd, NIST SRM 981 for Pb) were periodically analyzed to correct instrumental drift and geochemical reference materials including USGS BCR-2, BHVO-2, AVG-2, and RGM-2 were used in quality control.

**Mg isotopic diffusion modeling**. For the basalt-peridotite or carbonatite-peridotite systems, the network formation is expected, because the dihedral angles of these melt-rock systems are <60°[56]. Thus, the isotope composition of low-degree partial melts will be easily buffered by the matrix peridotites. In this scenario, the diffusion of Mg between the melts and matrix would be rate-limited by the diffusion in solid minerals. Considering that the diffusion rate of Mg in the major silicates (olivine, opx, garnet) is of the order of $10^{-16}$ m²/s[51], in the time scales of 0.1–100 Ma, the diffusion distance would be 0.02–0.56 m ($d = \sqrt{Dt}$), exceeding the typical grain sizes (<0.5 cm) of the asthenosphere. This means that the Mg isotopes of the low-degree partial melts would be totally reset during their percolation through the upper mantle. For the case of enriched elongate solid pyroxenite blobs within peridotitic mantle, we use the following equation to model the equilibrium between the blobs and matrix peridotites:

$$\delta^{26}Mg_{blobs}(t) = \delta^{26}Mg_{blobs} + \left(\delta^{26}Mg_{blobs(i)-\delta^{26}Mg_{blobs,eqm}}\right) \times erf\left(\frac{r}{\sqrt[2]{Dt}}\right)$$

where $\delta^{26}Mg_{blobs,eqm}$ is the $\delta^{26}Mg$ value of the pyroxenite in equilibrium with the peridotite matrix for a given temperature, $\delta^{26}Mg_{blobs}(i)$ is the assumed $\delta^{26}Mg$ value of the recycled eclogite/pyroxenite with a Mg isotopic anomaly, $r$ is the radius of blobs (in m), $t$ is time (s), and $D$ is the Mg diffusion coefficient (m²/s). $\delta^{26}Mg_{blobs,eqm}$ was determined by calculating $\delta^{26}Mg_{peridotite-pyroxenite}$ with the fractionation factors at 1350 °C[57].

## Data availability
All the data used in this study are reported in Supplementary Data 1–4.

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

## Acknowledgements
We thank F. Huang, H.M Yu, and Q.Z. Jin in USTC for their help in Mg isotope analysis. Jannick Ingrin and Eero Hanski are thanked for their help in the polishing the English and comments to the early version of manuscript. The discussion with Marc Hirschman also benefited to clarify some confusions. This study was supported by the Strategic Priority Research Program (B) of Chinese Academy of Sciences (grant No. XDB18000000), the National Natural Science Foundation of China (grant No. 91858214), the Scientific Research Fund of the Second Institute of Oceanography, MNR (grant No. HYGG 1801), National Key R&D Program of China (No. 2018YFC0309901), and the Toray Science and Technology Grant (#11–5208) and the Japan Society for the Promotion of Science (#24654180) to N.H.

## Author contributions
J.L. designed and led the project. N.H. led the cruise and provided the samples. J.L. conducted the Mg isotopic analysis, and wrote the paper with inputs from O.H., S.M, Q. K.X., and C.H.T. S.L.L, J.Liang, and T.D. prepared the samples and conducted the trace element and Sr–Nd–Pb isotope analyses. W.L. and W.F.Y. did the Mg isotopic diffusion modeling. G.Y.Z. helped to draw Fig. 1. All authors participated in the discussion and interpretations of results and in the preparation of the paper.

## Competing interests
The authors declare no competing interests.
