## [Peer Review File · Nature Communications]

Reviewers' comments:

Reviewer #1 (Remarks to the Author):

This is a very interesting paper with some excellent data and isotope-trace/major element correlations which I believe are novel. The overall story is potentially interesting and important, but I think it can be improved with consideration of the points below. I must admit, I found some of the arguments difficult to follow, not because they are necessarily poorly argued, but the story is quite complex. However, I would like to see this published - I think it's a novel and important contribution which many petrologists, geochemists and geophysicists will be very interested in.

A few more specific points are outlined below.

Line 53 - Sr-Nd-Pb isotopes presumably

Line 33-40 "Indeed, although volatile-rich silicate and carbonatite melts are thermodynamically stable beneath an old oceanic lithosphere, the maximum melt fraction is still more than one magnitude lower than the required melt fraction that can explain the occurrence of the G discontinuity, unless the melt fraction is efficiently enhanced by accumulation of deep-mantle-derived carbonatite/carbonated silicate melts or by a melting anomaly associated with an extra-volatile flux or the presence of a pyroxene-rich domain."

Please clarify this statement. At a fixed PT at any particular point on the LAB, the maximum melt fraction is controlled by the amount and nature of volatiles present, for a fixed bulk composition (i.e. peridotite). And how do melts accumulate? They are "upwardly mobile" even at small melt fractions, particularly in the case of carbonatite melts.

This paragraph needs clarification by referring to the well-known, experimentally-derived phase relations of peridotite + volatile melting at these conditions. Otherwise, correlating the G-discontinuity with the deep carbon cycle is rather tenuous at this stage.

Line 43 - please define "RF" for the non-seismologists

Figure 2 - which carbonatite was used on these plots to illustrate mixing? This is important as crustally emplaced carbonatites may not be compositionally the same as putative carbonatites in the mantle.

Lines 96-98 - I don't find this a particularly convincing argument. You could also look for correlations or no correlations between fluid mobile trace elements and Mg-isotopes, for example.

Figure 4 - I find the claimed correlations between Mg-isotope composition and various chemical compositional parameters to be rather unconvincing. They seem to be controlled by one or two of the low $\delta^{18}\text{O}$ samples being distinct from the rest, which are mostly within error of normal mantle values. The R-values are also low. Therefore, I am not convinced that these reflect a mixing process. Or am I missing the point here. I find the arguments about the process leading to the low Mg isotopes rather difficult to follow.

Paragraph beginning line 186 - this is the crux of the matter, if I understand correctly. The Mg-isotopes and trace elements can be decoupled by porous flow of carbonate melts through peridotite matrix. This is a neat idea.

Finally, I think the link between the geochemical observations (which are very interesting and informative in relation to potential sources of the magmas) and the cause of the G-discontinuity to be not well made. More detailed discussion of the amount and distribution of melts required to cause the seismic velocity drop is needed, and this needs to be better linked to the geochemistry

and the petrology (i.e. phase relations).

Reviewer #2 (Remarks to the Author):

The Gutenberg discontinuity is a seismic velocity discontinuity, often associated with the lithosphere asthenosphere boundary beneath oceanic plates. Its origin and, the physical and chemical properties that define them have remained enigmatic. Presence of partial melt is often invoked as one of the mechanisms to explain the sharp seismic velocity contrast, although the factors affecting melt production beneath older oceans is debated. This paper reports evidence that the melt fractions are enhanced by partial melts associated with recycling of ancient marine sediments and oceanic eclogite/pyroxenites. The authors arrive at this conclusion by studying Mg isotope data of basalts erupted on the northwest Pacific plate (petit spot lavas). The basis of their conclusion that these melts are responsible for the G discontinuity is derived from the observation of a sharp velocity discontinuity by Kawakatsu et al., (2009) on the ~131 Ma year old Pacific plate, close to their study/sampling regions.

Previous seismic studies of the Pacific Lithosphere have argued for the presence of melt closer to the ridges and regions altered by volcanism to explain the G discontinuity (Tharimena et al., 2017; Rychert and Shearer, 2011; Schmerr, 2012). Since mantle melt production is expected to be lower beneath older lithosphere, a more viable explanation for bulk of the observations of the G discontinuity beneath the Pacific is presence of frozen in melt. In regions of local enrichment such as mantle plumes or subducting slabs, volatiles could stabilize melt accumulation beneath older lithosphere (Sifre et al, 2014). In fact, Tharimena et al. (2017) argued for the presence of carbonatitic melt to explain some of their observations of sharp velocity change beneath older seafloor.

The present study provides further support and evidence to the notion of melt beneath older seafloor and the authors also provide a suitable mechanism to sustain the supply of melt through recycling of oceanic crustal material. This study provides a significant geochemical observation/constraint for seismic discontinuity observations in the older Pacific ocean closer to regions of subduction. As this paper provides important constraints and implications for future observations of the seismic G discontinuity globally, I can recommend this study for publication in Nature Communications provided the authors address these minor comments:

Minor Comments:

1) The base of the plate that translates from a rigid lithosphere to the mobile asthenosphere is called the lithosphere – asthenosphere boundary. Multiple factors influence this boundary including temperature, chemical composition, grain size, anisotropy and even extent of partial melt. While the G discontinuity is the top of the low velocity zone in the upper mantle (Revenaugh and Jordan, 1991). Since the G discontinuity coincides (in some oceanic regions) with the expected LAB depth, it is suggested that G discontinuity is closely related to the LAB.

My bone of contention (although minor) is that the authors use the terms LAB and G discontinuity interchangeably. This may be true for their study as presence of melt would weaken the mantle and therefore imply that the seismic G discontinuity defines the LAB in this region. However, beneath older lithosphere where the discontinuity in seismic velocity is due to say frozen in melt, then the G discontinuity will not coincide with the LAB.

I think the authors should make this distinction in the introduction and conclude the paper with the implication that the G discontinuity in their study region defines the LAB.

2) Fig 4: the text labels/annotations and legend are not scaled properly and are a bit hard to read. I would suggest using similar scaling as in Fig 2.

References:

Kawakatsu et al, 2009; Seismic evidence for sharp lithosphere-asthenosphere boundaries of

oceanic plates. *Science*, 324

Tharimena et al., 2017; Imaging Pacific lithosphere seismic discontinuities – Insights from SS precursor modeling. *JGR Solid Earth*, 122

Rychert and Shearer, 2011; Imaging the lithosphere – asthenosphere boundary beneath the Pacific using SS waveform modeling., *JGR*, 116

Schmerr, 2012; The Gutenberg discontinuity: Melt at the lithosphere – asthenosphere boundary, *Science*, 335

Sifre et al, 2014; Electrical conductivity during incipient melting in the oceanic low-velocity zone, *Nature*, 509

Revenaugh and Jordan, 1991; Mantle layering from ScS reverberations: 3. The upper mantle. *JGR*, 96

Reviewer #3 (Remarks to the Author):

Let me start by saying that I am not a high-temperature expert, and was asked to review this primarily for the methods. I make a number comments on these, but this is mainly in the form of tidying up and providing more information.

I have some suggestions for the main manuscript, which largely focus on examining the low-temperature effects more (dolomites and seawater), and also examining diffusion, which to my knowledge has been shown can yield more fractionation than you say.

Line 61-63: except of course by diffusion processes, which can generate significant fractionation. See Pogge von Strandmann et al., 2011, 2015; Lai et al., 2015; Hin et al. 2017.

Line 86: it's only really semantics, but the mean d26Mg values have evolved very slightly, with smaller uncertainty, in the past 9 years. See for example Lai et al., 2015 and Hin et al., 2017.

Line 88: "normal" mantle? Primitive mantle?

Line 96-97: I'm not convinced by this – there almost looks like there is a trend.

Line 103-109: I think this could be explored a bit more, especially in the context of potential dolomites.

Line 112: add 'g'

Line 193-200: see Pogge von Strandmann et al 2015 where scenarios are explored where an integrated pore fluid network causes diffusional heterogeneity of over 1‰.

Methods: given that this is an NComms paper, I think there needs to be more description of the method. So, for example, give numbers for the columns yields. The d26Mg values for the USGS standards should definitely be given, as should be their reproducibility. At the moment you only provide information for BHVO-2 and BCR-2, and not any of the other standards you mention. This should also include the 'in-house' standards, because, at least in the case of CAM-1 and SRM980, are not actual 'in-house' standards, but have been measured by a large number of groups.

I would also like to know exactly what the N means in your Supp. Table 4: is that for complete reanalyses (i.e. including dissolution and chemistry), or analysis repeats during a single run, or what?

I would also suggest that you expand on the 'recommended' values for BHVO-2 and BCR-2. The number of An et al are quite heavy compared to many other studies, especially in light of the precision you are reporting. You could look at the compilation in Pogge von Strandmann et al.,

2011, or indeed a swift perusal of GeoRem shows that there are 38 different publications for $\delta^{26}\text{Mg}$ for BCR-2 and 31 for BHVO-2.

Lastly, you don't seem to have repeated any of the standards (?), so it is difficult to provide a true test of combined accuracy and precision.

Reviewers' comments:

Reviewer #1 (Remarks to the Author):

This is a very interesting paper with some excellent data and isotope-trace/major element correlations which I believe are novel. The overall story is potentially interesting and important, but I think it can be improved with consideration of the points below. I must admit, I found some of the arguments difficult to follow, not because they are necessarily poorly argued, but the story is quite complex. However, I would like to see this published - I think it's a novel and important contribution which many petrologists, geochemists and geophysicists will be very interested in.

A few more specific points are outlined below.

Line 53 - Sr-Nd-Pb isotopes presumably

It has been revised.

Line 33-40 "Indeed, although volatile-rich silicate and carbonatite melts are thermodynamically stable beneath an old oceanic lithosphere, the maximum melt fraction is still more than one magnitude lower than the required melt fraction that can explain the occurrence of the G discontinuity, unless the melt fraction is efficiently enhanced by accumulation of deep-mantle-derived carbonatite/carbonated silicate melts or by a melting anomaly associated with an extra-volatile flux or the presence of a pyroxene-rich domain."

Please clarify this statement. At a fixed PT at any particular point on the LAB, the maximum melt fraction is controlled by the amount and nature of volatiles present, for a fixed bulk composition (i.e. peridotite). And how do melts accumulate? They are "upwardly mobile" even at small melt fractions, particularly in the case of carbonatite melts.

This paragraph needs clarification by referring to the well-known, experimentally-derived phase relations of peridotite + volatile melting at these conditions. Otherwise, correlating the G-discontinuity with the deep carbon cycle is rather tenuous at this stage.

We are sorry to induce the confusion. We intended to introduce the conundrum to the readers, that is on one hand the sharp seismic velocity decrease at LAB (G discontinuity) require melt with fraction at least at 0.25 to 1.25%, on the other hand the mantle with normal amount of volatiles (content like the MORB source) only allow partial melting of the degree one order of magnitude lower. Then, we pointed out that in order to explain G discontinuity with presence of melt, additional processes/mechanisms enhancing the melt proportion should have played their roles. Thus, we did not intend to correlate the G with deep carbon cycle here.

We agree with you that for a fixed P-T condition and a fixed bulk composition, the maximum melt fraction would be controlled by the amount and nature of volatiles, in the pure view of thermodynamic equilibrium. However, the melts in the mantle could be locally accumulated due to several factors. The first one is due to the deformation of the mantle (Holzman et al., 2003, Science; Katz et al., 2006, Nature). The second one is the density differences (Sakamaki et al., 2013). The lower density of the carbonated silicates or carbonatite melts in the upper mantle would make them to migrate upwards in the regions of decompressing mantle, and become trapped at the base of the LAB. Because the LAB is a freezing front, it would be quasi-impermeable to further migration of the melts (both silicate or carbonatite).

The partial melting degree of the mantle induced by the presence of volatile with CO₂ and H₂O content in the range of the N-MORB like mantle source, at the normal P-T condition beneath the old oceanic lithosphere would be much lower than 1%. This percentage of partial melting degree is significantly lower than that present in the partial melting experiments, and in these experiments the initial CO₂ or H₂O content were usually much higher than the practical content the normal mantle could have. Thus, it is not meaningful here to relate the G to these experiments here. Actually, in the calculation of the distribution of melt in the LVZ as a function of depth and age of overlying lithosphere of mantle with the H₂O and CO₂ content of MORB source, Hirschman (2010, PEPI, what we have cited in our manuscript) has accounted for the previous partial melting experiments on the CO₂-bearing peridotite system with pressure from 1.8GPa to 8 GPa (Dasgupta et al., 2007, *Geology*; Gudfinnsson and Presnall, 2005), and on the H₂O-bearing peridotite system (parameterized in Hirschman et al., 2009). Thus, we still prefer not to infer those experiments directly in our manuscript.

However, we tried to add more detail in this part (see line 31 to 41), to make it easier to understand.

Line 43 - please define "RF" for the non-seismologists

It represent receiver function. We have defined this in the revised manuscript (line 46 in the new manuscript).

Figure 2 - which carbonatite was used on these plots to illustrate mixing? This is important as crustally emplaced carbonatites may not be compositionally the same as putative carbonatites in the mantle.

We used the average value of the reported carbonatites from young (<130 Ma) volcanics associated with the East African Rift and the Canadian Craton and from Proterozoic complexes from the South African craton, which are reported by Bizimis et al. (2003, *CMP*).

Yes, the erupted or intrusive carbonatites are believed to represent either primary melts from the mantle or products of liquid immiscibility/crystal fractionation. For these carbonatites, the carbonate and non-carbonate fractions show obvious Nd-Sr-Hf isotopes disequilibrium, which roles out the origin of non-carbonates as the products of immiscibility/crystal fractional from an evolving carbonatitic magma. Instead, the non-carbonate parts would be the xenocrysts. On the other hand, the modeled trace element pattern of a primary carbonatite melt derived from the carbonated peridotite is rather consistent with the pattern of the natural carbonatite, especially the depletion of Zr, Hf, and Ti relative to the nearby elements (Dasgupta et al., 2009). Thirdly, the experiments on the nephelinite-dolomite-NaCO₃ joint (Lee and Wyllie, 1997) crystallized olivine, clinopyroxene, plus traces of spinel along with quenched silicate and carbonate liquids and calcite. For the Mg-free system Na-Al-Ca-Si+CO₂, the melilite, anorthite, nepheline and scapolite crystallized. Fractionation of these minerals would not produce the extreme LREE/HREE ratios, and with high Zr/Hf and Nb/Ta ratios, without the progenitor carbonatite melt being already highly enriched in these elements.

Overall, we followed the conclusions that these erupted carbonatites could represent the primary carbonatite from the mantle. In the revised manuscript, we adopted the average composition of the carbonate parts, in order to exclude/reduce the possible effects of the silicate parts. As shown in the new figure, this does not change our previous conclusion.

Lines 96-98 - I don't find this a particularly convincing argument. You could also look for correlations or no correlations between fluid mobile trace elements and Mg-isotopes, for example. Yes, we have reconsidered this argument in the revised version (line 103-105 in the new manuscript). Actually, whether there is or is not a correlation (positive, if any) between LOI and deltaMg, the clear truth is the lowest delta26Mg is preserved in the freshest sample, and there is no negative correlation between LOI and delta26Mg (which should be expected if alteration play significant role in modifying the delta26Mg). Thus, the identification of this correlation does not change our conclusion.

Figure 4 - I find the claimed correlations between Mg-isotope composition and various chemical compositional parameters to be rather unconvincing. They seem to be controlled by one or two of the low delMg samples being distinct from the rest, which are mostly within error of normal mantle values. The R-values are also low. Therefore, I am not convinced that these reflect a mixing process. Or am I missing the point here. I find the arguments about the process leading to the low Mg isotopes rather difficult to follow.

Yes, the R-values are relatively low, but if you use T-test to check these trends you will find these correlations are significant in the 95% confidence range. Actually, the correlations among Mg isotopes and Pb isotopes and trace elemental ratios are inter-consistent. For instance, delta26Mg, Ce/Pb and $^{206}\text{Pb}/^{204}\text{Pb}$ ratios are mutually correlated, as shown in Figure 2 and Figure 4. The multiple linear correlations among Pb isotopes and many trace elemental ratios (shown in Figure 2) are at best explained as the mixing of magma with two end members. This should be the base on which we consider the Mg isotopes. We agree with you that the Mg isotopes are relatively scattered, but when all the data were combined together, it makes sense.

About the origin of the low deltaMg. The logic is here. After excluding the possible role from the shallow magma process and the post alteration, we have attributed the low delta26Mg to the mantle source heterogeneity. The possible candidates for sub-normal mantle delta26Mg is recycled oceanic carbonate or the eclogite/pyroxenites that transformed from the carbonated basaltic oceanic crust (Hu Yan et al., 2018; Wang et al., 2017). These kinds of eclogites/pyroxenites would heritage the low deltaMg signature through reaction of Mg-bearing carbonate (low deltaMg hosting) with mafic minerals in the oceanic crust during transport into the deep mantle, but do not necessarily still contain carbonate when they go back to the upper mantle after the long journey of recycling. As we shown by the multiple lines of correlations in Figure 2 and Figure 4, we have suggested two end members, with high $^{206}\text{Pb}/^{204}\text{Pb}$ but low delta26Mg, and low $^{206}\text{Pb}/^{204}\text{Pb}$ but high delta26Mg, respectively. The low delta26Mg components show high Ti/Eu, low Zr/Hf and Nb/Ta ratios that are usually contrast to the carbonatite. So, we suggest that the low deltaMg would come from the recycled eclogites/pyroxenites who get their low deltaMg during their processor's subducting.

Paragraph beginning line 186 - this is the crux of the matter, if I understand correctly. The Mg-isotopes and trace elements can be decoupled by porous flow of carbonate melts through peridotite matrix. This is a neat idea.

Finally, I think the link between the geochemical observations (which are very interesting and informative in relation to potential sources of the magmas) and the cause of the G-discontinuity to be not well made. More detailed discussion of the amount and distribution of melts required to cause the seismic velocity drop is needed, and this needs to be better linked to the geochemistry and the petrology (i.e. phase relations).

Thanks for your suggestions. Frankly speaking, when we organized the first version of this manuscript, we also tried to make some quantitative constraints on the linkage between G-discontinuity to the recycled CO₂ and pyroxenite in the source. However, we found it is truly hard to figure out this issue on the basis of our isotopes and trace-elements data set. First of all, even the estimated partial melt fraction required to account for the observed shear-wave velocity drop spans a rather large range. As shown in the early seismic work by Kawakatsu et al. (2009, Science), the estimated melt fraction would be largely depending on how the mantle in the LAB melted. The two end member scenarios are the homogenous, equilibrated partially molten, and the horizontal melt layer. For the former case, the melt fraction should be ~3.5 to 4.0%, while the latter case just requires 0.25 to 1.5%. The real situation would be between these two end members. Secondly, although we identified two end members components (one is related to carbonate, and the other is related to pyroxenite) in the source, it is hard to estimate the proportions of the melts with these two kinds of geochemical signatures within the LAB, because multiple processes involved in the melt transportation and storage after melt extraction from mantle would disproportionally sample those two kinds of melts. Thirdly, the partial melting degree of the pyroxenites/eclogite/cpx enriched region in the mantle at a specific P-T conditions, would largely depend on the bulk composition (Mg#, total alkali, Si-rich or Si-poor). These conditions are actually not easy/possible to identify by our isotopes and trace-elements data set.

Actually, the freshest and important contribution of our paper is that, we firstly, give clear geochemical evidences to the styles/approaches of melt fraction enhancement that responsible for the G. The quantitative constraints would be done in the future work, if we can account for the problems well.

Reviewer #2 (Remarks to the Author):

The Gutenberg discontinuity is a seismic velocity discontinuity, often associated with the lithosphere asthenosphere boundary beneath oceanic plates. Its origin and, the physical and chemical properties that define them have remained enigmatic. Presence of partial melt is often invoked as one of the mechanisms to explain the sharp seismic velocity contrast, although the factors affecting melt production beneath older oceans is debated. This paper reports evidence that the melt fractions are enhanced by partial melts associated with recycling of ancient marine sediments and oceanic eclogite/pyroxenites. The authors arrive at this conclusion by studying Mg isotope data of basalts erupted on the northwest Pacific plate (petit spot lavas). The basis of their conclusion that these melts are responsible for the G discontinuity is derived from the observation of a sharp velocity discontinuity by Kawakatsu et al., (2009) on the ~131 Ma year old Pacific plate, close to their study/sampling regions.

Previous seismic studies of the Pacific Lithosphere have argued for the presence of melt closer to the ridges and regions altered by volcanism to explain the G discontinuity (Tharimena et al., 2017; Rychert and Shearer, 2011; Schmerr, 2012). Since mantle melt production is expected to be lower beneath older lithosphere, a more viable explanation for bulk of the observations of the G discontinuity beneath the Pacific is presence of frozen in melt. In regions of local enrichment such as mantle plumes or subducting slabs, volatiles could stabilize melt accumulation beneath older lithosphere (Sifre et al, 2014). In fact, Tharimena et al. (2017) argued for the presence of

carbonatitic melt to explain some of their observations of sharp velocity change beneath older seafloor.

The present study provides further support and evidence to the notion of melt beneath older seafloor and the authors also provide a suitable mechanism to sustain the supply of melt through recycling of oceanic crustal material. This study provides a significant geochemical observation/constraint for seismic discontinuity observations in the older Pacific ocean closer to regions of subduction. As this paper provides important constraints and implications for future observations of the seismic G discontinuity globally, I can recommend this study for publication in Nature Communications provided the authors address these minor comments:

Minor Comments:

1) The base of the plate that translates from a rigid lithosphere to the mobile asthenosphere is called the lithosphere – asthenosphere boundary. Multiple factors influence this boundary including temperature, chemical composition, grain size, anisotropy and even extent of partial melt. While the G discontinuity is the top of the low velocity zone in the upper mantle (Revenaugh and Jordan, 1991). Since the G discontinuity coincides (in some oceanic regions) with the expected LAB depth, it is suggested that G discontinuity is closely related to the LAB.

My bone of contention (although minor) is that the authors use the terms LAB and G discontinuity interchangeably. This may be true for their study as presence of melt would weaken the mantle and therefore imply that the seismic G discontinuity defines the LAB in this region. However, beneath older lithosphere where the discontinuity in seismic velocity is due to say frozen in melt, then the G discontinuity will not coincide with the LAB.

I think the authors should make this distinction in the introduction and conclude the paper with the implication that the G discontinuity in their study region defines the LAB.

Thanks for your suggestions, we have clarify this in the revised manuscript. Actually, as the conclusion we shown in the abstract, we really focused on the G discontinuity beneath the normal oceanic regions. See line 25 to 28 in the new manuscript.

2) Fig 4: the text labels/annotations and legend are not scaled properly and are a bit hard to read. I would suggest using similar scaling as in Fig 2.

This has been revised.

References:

Kawakatsu et al, 2009; Seismic evidence for sharp lithosphere-asthenosphere boundaries of oceanic plates. Science, 324

Tharimena et al., 2017; Imaging Pacific lithosphere seismic discontinuities – Insights from SS precursor modeling. JGR Solid Earth, 122

Rychert and Shearer, 2011; Imaging the lithosphere – asthenosphere boundary beneath the Pacific using SS waveform modeling., JGR, 116

Schmerr, 2012; The Gutenberg discontinuity: Melt at the lithosphere – asthenosphere boundary, Science, 335

Sifre et al, 2014; Electrical conductivity during incipient melting in the oceanic low-velocity zone, Nature, 509

Revenaugh and Jordan, 1991; Mantle layering from ScS reverberations: 3. The upper mantle. JGR, 96

Reviewer #3 (Remarks to the Author):

Let me start by saying that I am not a high-temperature expert, and was asked to review this primarily for the methods. I make a number comments on these, but this is mainly in the form of tidying up and providing more information.

I have some suggestions for the main manuscript, which largely focus on examining the low-temperature effects more (dolomites and seawater), and also examining diffusion, which to my knowledge has been shown can yield more fractionation than you say.

Thanks for your suggestions. We have considered the role of dolomites and seawater in the revised manuscript. However, we still hold our previous view that at high temperature, the diffusion during the transport of parous flow will not explain our data. Please see the explanation in the following answers for your comments on Lines 193-200.

Line 61-63: except of course by diffusion processes, which can generate significant fractionation. See Pogge von Strandmann et al., 2011, 2015; Lai et al., 2015; Hin et al. 2017.

Thanks for your suggestions. Yes, the diffusion processes could lead to larger extend of Mg isotope fractions. However, the reported diffusion-inducing Mg isotopes fractionations in the literatures so far were rightly for the case of metamorphism at low temperature and fluids-activity related (Pogge von Strandmann et al., 2015, ~450°C), or for the interaction between xenocrysts or phenocrysts and hosting magma (Sio et al., 2017; Oeser et al., 2015). These situations are rather different from the processes of mantle partial melting, melt segregation in the mantle, and its fractionation during the ascent. To our knowledge until now there is no case of nature basalt, large amount of bulk rock, were explained as the result of diffusion for their Mg isotope variations.

For Pogge von Strandmann et al. (2011), the author reported the global xenoliths from different background, from craton, off craton to arc. The co-variation of delta Li and deltaMg would most likely reflect the metasomatism by different agency. It would be hard to consider one simple diffusion process to lead to the globally variation.

For Pogge von Strandmann et al. (2015), see the explanation in the following part.

Lai et al. (2015): While the Li isotopes span large range, the Homoran peridotites show rather limited range of deltaMg. The authors concluded that “The constant d26Mg values of Horoman peridotites suggest that chemical potential gradients caused by melt infiltration were insufficient to drive associated d26Mg fractionation greater than our external precision of 0.03‰”.

Finally, the paper by Hin et al. (2017, Nature) reported the fraction of Mg isotopes not by diffusion, but by the accretional vapour loss.

Anyway, we add the reference of Pogge von Strandmann et al. (2015), which should be covered by low-temperature processes.

Line 86: it's only really semantics, but the mean d26Mg values have evolved very slightly, with smaller uncertainty, in the past 9 years. See for example Lai et al., 2015 and Hin et al., 2017.

Thanks for your reminding. The range suggested by these two references are slightly heavier than the value by Teng et al. (2010). This would not change our discussion and conclusions.

Line 88: “normal” mantle? Primitive mantle?

We are sorry to cause the confusion. We intended to compare the Site B samples with the global MORB and OIB source. We have clarified this in the revised version.

Line 96-97: I'm not convinced by this – there almost looks like there is a trend.

Yes, it is not so clear on megascopic, and the R^2 of the linear fitting is relatively low (0.36). But this does not change our conclusion that the alteration is the not reason for the low $\delta^{26}\text{Mg}$. The important thing is that the low $\delta^{26}\text{Mg}$ is preserved in the sample with least alteration. In the revised manuscript, we do not mention a “positive trend”.

Line 103-109: I think this could be explored a bit more, especially in the context of potential dolomites.

Actually, the marine sediments with low $\delta^{26}\text{Mg}$ are mainly due to the precipitation of dolomite (Hu et al., 2017). Such low $\delta^{26}\text{Mg}$ sediments have high radiogenic Pb isotopes (Hu et al., 2017) and low Ce/Pb ratios, thus the assimilation of such sediments could be largely identified by the correlations of Pb isotope and trace element ratios (revised Figure 1).

Yes, in the revised manuscript, we have considered the possible effect of dolomites.

Line 112: add ‘g’

This has been revised according to your suggestion (see Line 122 in the revised manuscript).

Line 193-200: see Pogge von Strandmann et al 2015 where scenarios are explored where an integrated pore fluid network causes diffusional heterogeneity of over 1%.

Thanks for your suggestions. I know this paper, which reported large Mg isotopic fractionation of the reaction front of an exhumed contact between rocks of subducted crust and serpentinite in the Syros melange zone. The low $\delta^{26}\text{Mg}$ were explained as the results of diffusion through an interconnected pore fluid network at $\sim 430^\circ\text{C}$. The authors suggest that “Grain boundary network diffusion would create an isotopically heterogeneous fluid that has light isotope ratios at the boundary of the minerals (dominantly chlorite) growing by diffusive supply of Mg”. The critical point for this model is that the authors did not consider the re-equilibrium between the parous fluids and the matrix body, which would buffer the Mg isotopic fractionation caused by diffusion, because at the such low temperature the diffusion in the solid is too sluggish. This solution would be reasonable for this case.

However, it could not be applied to our case because of the following reasons. The high ambient mantle temperature ($\sim 1350^\circ\text{C}$) would result in: 1) the fast diffusion within the minerals, which would reset the Mg isotopic fractionation of the parous flow caused by diffusion during their transport; 2) extensive the reaction of the melt with the mantle rock during the melt (initially it is carbonatite in our case) transport, which transfer the carbonatite to carbonated silicates. Due to the two processes above, the Mg isotopic composition of the parous flow in our case would be buffered by the ambient mantle largely. This would be the partial reason why globally the $\delta^{26}\text{Mg}$ of both MORB and OIB are consistent with the mantle peridotite (except the source heterogeneity for few cases, like Pitcairn EM1 basalts, Wang et al., 2019).

Thus, we still hold the previous view that during the melt percolation within the asthenosphere, no significant Mg isotopic fractionation could be caused by diffusion. To the contrary, we suggest it is the buffering by the ambient mantle, the low $\delta^{26}\text{Mg}$ of the carbonatite melt inherited from the mantle source was erased.

Methods: given that this is an NComms paper, I think there needs to be more description of the method. So, for example, give numbers for the columns yields. The $\delta^{26}\text{Mg}$ values for the USGS standards should definitely be given, as should be their reproducibility. At the moment you only provide information for BHVO-2 and BCR-2, and not any of the other standards you mention.

This should also include the ‘in-house’ standards, because, at least in the case of CAM-1 and SRM980, are not actual ‘in-house’ standards, but have been measured by a large number of groups.

I would also like to know exactly what the N means in your Supp. Table 4: is that for complete reanalyses (i.e. including dissolution and chemistry), or analysis repeats during a single run, or what?

The column yields are always better than 99.5%.

We are sorry in the previous version we made a mistake on the description of the Mg isotope method. The approach reported by An et al. (2014) conducted many analyses on international standards, that is in order to build the standard procedure and compare with other labs. For our samples, at 2017, the method in this lab has been routine, so we did not actually use so many standards. In our analysis session, we used the BHVO-2 and BCR-2 as the standards to see whether our dissolution, column chemistry and MS analysis could produce results comparable with other international labs (accuracy), and used in-house standard (IGG) bracketed the unknown samples to see the repeatability. We have added the repeated analysis on IGG in Supplementary Table 4.

In this table, n represent the times of repeat analysis by MS for the same Mg purified cuts, which is mentioned in the note of table. Actually, we also have marked the complete re-analyses including dissolution and chemistry by “replicate” in this table.

I would also suggest that you expand on the ‘recommended’ values for BHVO-2 and BCR-2. The number of An et al are quite heavy compared to many other studies, especially in light of the precision you are reporting. You could look at the compilation in Pogge von Strandmann et al., 2011, or indeed a swift perusal of GeoRem shows that there are 38 different publications for d26Mg for BCR-2 and 31 for BHVO-2.

Thanks for your suggestions. We have expanded the recommended values for these two standards in Table 4.

We checked the recent compilation of analysis on Mg isotope standards by Teng (2017, RMG, Table 1). We see for the BHVO-2, the analyzed value by Pogge von Strandmann et al. (2008, 2011) is $-0.25 \pm 0.11\%$, and $-0.26 \pm 0.06\%$, respectively, while the recommended value by Teng is $-0.24 \pm 0.08\%$. These values are comparable with An’s recommended value for this standard ($-0.22 \pm 0.04\%$). For BCR-2, the analyzed values from different labs vary from $-0.16 \pm 0.01\%$ to $-0.33 \pm 0.04\%$. Yes, An’s value is heavier than the high limit value, but is still consistent with $-0.16 \pm 0.11\%$ (Bourdon et al., 2010), -0.19 ± 0.02 (Baker et al., 2005), $-0.17 \pm 0.35\%$ (Bizzarro et al., 2004), $-0.14 \pm 0.11\%$ (Wombacher et al., 2009), -0.16 ± 0.11 (Tipper et al., 2008), $-0.25 \pm 0.06\%$ (Pogge von Strandmann et al., 2011), and $-0.20 \pm 0.07\%$ (Wimpenny et al., 2014b). As suggested by An et al. (2014), the discrepancy with the lighter values (< -0.3) would be possibly due to the “dry” plasma was used for the latter. Anyway, during our session, the analyzed BCR-2 value is $-0.22 \pm 0.06\%$, which is consistent with many published values within errors.

Lastly, you don’t seem to have repeated any of the standards (?), so it is difficult to provide a true test of combined accuracy and precision.

Actually, during our analysis session, the international standards were analyzed and compared with the recommended value. If the measured value for these standards (BCR-2, BHVO-2) are consistent with the recommended value within the uncertainty (2sd of multiple analysis for the same purified cuts), we say our results should be “accuracy”, and comparable crossing the labs.

The precision could be seen by the repeated analysis of the in-house standards and selected samples (repeat here means complete re-do of dissolution, column chemistry, and analysis by MS). We have added the repeated analysis of in house standards in the Supplementary Table 4.

Reviewers' comments:

Reviewer #1 (Remarks to the Author):

This is a very interesting paper with some excellent data and isotope-trace/major element correlations which I believe are novel. The overall story is potentially interesting and important, but I think it can be improved with consideration of the points below. I must admit, I found some of the arguments difficult to follow, not because they are necessarily poorly argued, but the story is quite complex. However, I would like to see this published - I think it's a novel and important contribution which many petrologists, geochemists and geophysicists will be very interested in.

A few more specific points are outlined below.

Line 53 - Sr-Nd-Pb isotopes presumably

We have replaced “isotope” with “isotopes”, as shown in Line 56 in the revised version.

Line 33-40 "Indeed, although volatile-rich silicate and carbonatite melts are thermodynamically stable beneath an old oceanic lithosphere, the maximum melt fraction is still more than one magnitude lower than the required melt fraction that can explain the occurrence of the G discontinuity, unless the melt fraction is efficiently enhanced by accumulation of deep-mantle-derived carbonatite/carbonated silicate melts or by a melting anomaly associated with an extra-volatile flux or the presence of a pyroxene-rich domain."

Please clarify this statement. At a fixed PT at any particular point on the LAB, the maximum melt fraction is controlled by the amount and nature of volatiles present, for a fixed bulk composition (i.e. peridotite). And how do melts accumulate? They are “upwardly mobile” even at small melt fractions, particularly in the case of carbonatite melts.

We are sorry to induce the confusion. We agree with you that for a fixed P-T condition and a fixed bulk composition, the maximum melt fraction would be controlled by the amount and nature of volatiles, in the pure view of thermodynamic equilibrium. However, the melts in the mantle could be locally accumulated due to several factors. The first one is due to the deformation of the mantle (Holzman et al., 2003, Science; Katz et al., 2006, Nature). The second one is the density

differences (Sakamaki et al., 2013). The lower density of the carbonated silicates or carbonatite melts in the upper mantle would make them to migrate upwards in the regions of decompressing mantle, and become trapped at the base of the LAB. Because the LAB is a freezing front, it would be quasi-impermeable to further migration of the melts (both silicate or carbonatite).

We have added these statement in the revised introduction (Line 37 to 41, in the revised manuscript).

This paragraph needs clarification by referring to the well-known, experimentally-derived phase relations of peridotite + volatile melting at these conditions. Otherwise, correlating the G-discontinuity with the deep carbon cycle is rather tenuous at this stage.

Actually, in the introduction part we intended to introduce the conundrum to the readers, that is on one hand the sharp seismic velocity decrease at LAB (G discontinuity) require melt with fraction at least at 0.25 to 1.25%, on the other hand the mantle with normal amount of volatiles (content like the MORB source) only allow partial melting of the degree one order of magnitude lower. Then, we pointed out that in order to explain G discontinuity with presence of melt, additional processes/mechanisms enhancing the melt proportion should have played their roles. Thus, we did not intend to correlate the G with deep carbon cycle here.

The partial melting degree of the mantle induced by the presence of volatile with CO₂ and H₂O content in the range of the N-MORB like mantle source, at the normal P-T condition beneath the old oceanic lithosphere would be much lower than 1%. This percentage of partial melting degree is significantly lower than that present in the partial melting experiments, and in these experiments the initial CO₂ or H₂O content were usually much higher than the practical content the normal mantle could have. Thus, we guess it is not very meaningful here to relate the G to these experiments directly. Actually, in the calculation of the distribution of melt in the LVZ as a function of depth and age of overlying lithosphere of mantle with the H₂O and CO₂ content of MORB source, Hirschman (2010, PEPI, what we have cited in our manuscript) has accounted for the previous partial melting experiments on the CO₂-bearing peridotite system with pressure from 1.8 GPa to 8 GPa (Dasgupta et al., 2007, *Geology*; Gudfinnsson and Presnall, 2005), and on the H₂O-bearing peridotite system (parameterized in Hirschman et al., 2009).

However, we have mentioned that the background of the calculation by Hirschman (2010), in which those CO₂-bearing partial melting experiments have been involved. See Line from 31 to 41

in the revised manuscript.

Line 43 - please define "RF" for the non-seismologists

It represent receiver function. We have defined this in the revised manuscript (line 46 in the new manuscript).

Figure 2 - which carbonatite was used on these plots to illustrate mixing? This is important as crustally emplaced carbonatites may not be compositionally the same as putative carbonatites in the mantle.

We used the average value of the reported carbonatites from young (<130 Ma) volcanics associated with the East African Rift and the Canadian Craton and from Proterozoic complexes from the South African craton, which are reported by Bizimis et al. (2003, CMP).

Yes, the erupted or intrusive carbonatites are believed to represent either primary melts from the mantle or products of liquid immiscibility/crystal fractionation. For these carbonatites, the carbonate and non-carbonate fractions show obvious Nd-Sr-Hf isotopes disequilibrium, which rules out the origin of non-carbonates as the products of immiscibility/crystal fractional from an evolving carbonatitic magma. Instead, the non-carbonate parts would be the xenocrysts. On the other hand, the modeled trace element pattern of a primary carbonatite melt derived from the carbonated peridotite is rather consistent with the pattern of the natural carbonatite, especially the depletion of Zr, Hf, and Ti relative to the nearby elements (Dasgupta et al., 2009). Thirdly, the experiments on the nephelinite-dolomite-NaCO₃ joint (Lee and Wyllie, 1997) crystallized olivine, clinopyroxene, plus traces of spinel along with quenched silicate and carbonate liquids and calcite. For the Mg-free system Na-Al-Ca-Si+CO₂, the melilite, anorthite, nepheline and scapolite crystallized. Fractionation of these minerals would not produce the extreme LREE/HREE ratios, and with high Zr/Hf and Nb/Ta ratios, without the progenitor carbonatite melt being already highly enriched in these elements.

Overall, we followed the conclusions that these erupted carbonatites could represent the primary carbonatite from the mantle. In the revised manuscript, we adopted the average composition of the carbonate parts, in order to exclude/reduce the possible effects of the silicate parts. See the new Figure 3 (c,d), we have revised the modeling calculation. This does not change our previous conclusion.

Lines 96-98 - I don't find this a particularly convincing argument. You could also look for

correlations or no correlations between fluid mobile trace elements and Mg-isotopes, for example.

Thanks for your reminding. Actually, whether there is or is not a correlation (positive, if any) between LOI and $\delta^{26}\text{Mg}$, the clear truth is the lowest $\delta^{26}\text{Mg}$ is preserved in the freshest sample, and there is no negative correlation between LOI and $\delta^{26}\text{Mg}$ (which should be expected if alteration play significant role in modifying the $\delta^{26}\text{Mg}$). Thus, the identification of this correlation does not change our conclusion.

We have reconsidered this argument in the revised version (line 103-105 in the new manuscript).

Figure 4 - I find the claimed correlations between Mg-isotope composition and various chemical compositional parameters to be rather unconvincing. They seem to be controlled by one or two of the low $\delta^{26}\text{Mg}$ samples being distinct from the rest, which are mostly within error of normal mantle values. The R-values are also low. Therefore, I am not convinced that these reflect a mixing process. Or am I missing the point here. I find the arguments about the process leading to the low Mg isotopes rather difficult to follow.

Yes, the R-values are relatively low, but if you use T-test to check these trends you will find these correlations are significant in the 95% confidence range. Actually, the correlations among Mg isotopes and Pb isotopes and trace elemental ratios are inter-consistent. For instance, $\delta^{26}\text{Mg}$, Ce/Pb and $^{206}\text{Pb}/^{204}\text{Pb}$ ratios are mutually correlated, as shown in Figure 2 and Figure 4. The multiple linear correlations among Pb isotopes and many trace elemental ratios (shown in Figure 2) are at best explained as the mixing of magma with two end members. This should be the base on which we consider the Mg isotopes (see line 141 to 145 in the revised manuscript). We agree with you that the Mg isotopes are relatively scattered, but when all the data were combined together, it makes sense. We add the some of these explanations to the revised manuscript (see line 123 to 124).

About the origin of the low $\delta^{26}\text{Mg}$. The logic is here. After excluding the possible role from the shallow magma process and the post alteration, we have attributed the low $\delta^{26}\text{Mg}$ to the mantle source heterogeneity. The possible candidates for sub-normal mantle $\delta^{26}\text{Mg}$ is recycled oceanic carbonate or the eclogite/pyroxenites that transformed from the carbonated basaltic oceanic crust (Hu Yan et al., 2018; Wang et al., 2017). These kinds of eclogites/pyroxenites would heritage the low $\delta^{26}\text{Mg}$ signature through reaction of Mg-bearing

carbonate (low $\delta^{26}\text{Mg}$ hosting) with mafic minerals in the oceanic crust during transport into the deep mantle, but do not necessarily still contain carbonate when they go back to the upper mantle after the long journey of recycling. As we shown by the multiple lines of correlations in Figure 2 and Figure 4, we have suggested two end members, with high $^{206}\text{Pb}/^{204}\text{Pb}$ but low $\delta^{26}\text{Mg}$, and low $^{206}\text{Pb}/^{204}\text{Pb}$ but high $\delta^{26}\text{Mg}$, respectively. The low $\delta^{26}\text{Mg}$ components show high Ti/Eu, low Zr/Hf and Nb/Ta ratios that are usually contrast to the carbonatite. So, we suggest that the low $\delta^{26}\text{Mg}$ would come from the recycled eclogites/pyroxenites who get their low $\delta^{26}\text{Mg}$ during their processor's subducting. Overall, we did not modify the relevant text in the revised manuscript.

Paragraph beginning line 186 - this is the crux of the matter, if I understand correctly. The Mg-isotopes and trace elements can be decoupled by porous flow of carbonate melts through peridotite matrix. This is a neat idea.

Finally, I think the link between the geochemical observations (which are very interesting and informative in relation to potential sources of the magmas) and the cause of the G-discontinuity to be not well made. More detailed discussion of the amount and distribution of melts required to cause the seismic velocity drop is needed, and this needs to be better linked to the geochemistry and the petrology (i.e. phase relations).

Thanks for your suggestions. Frankly speaking, when we organized the first version of this manuscript, we also tried to make some quantitatively constraints on the linkage between G-discontinuity to the recycled CO_2 and pyroxenite in the source. However, we found it is truly hard to figure out this issue on the basis of our isotopes and trace-elements data set. First of all, even the estimated partial melt fraction required to account for the observed shear-wave velocity drop spans a rather large range. As shown in the early seismic work by Kawakatsu et al. (2009, Science), the estimated melt fraction would be largely depending on how the mantle in the LAB melted. The two end member scenarios are the homogenous, equilibrated partially molten, and the horizontal melt layer. For the former case, the melt fraction should be ~ 3.5 to 4.0%, while the latter case just requires 0.25 to 1.5%. The real situation would be between these two end members. Secondly, although we identified two end members components (one is related to carbonate, and the other is related to pyroxenite) in the source, it is hard to estimate the proportions of the melts

with these two kinds of geochemical signatures within the LAB, because multiple processes involved in the melt transportation and storage after melt extraction from mantle would disproportionally sample those two kinds of melts. Thirdly, the partial melting degree of the pyroxenites/eclogite/cpx enriched region in the mantle at a specific P-T conditions, would largely depend on the bulk composition (Mg#, total alkali, Si-rich or Si-poor). These conditions are actually not easy/possible to identify by our isotopes and trace-elements data set.

Actually, the freshest and important contribution of our paper is that, we firstly, give clear geochemical evidences to the styles/approaches of melt fraction enhancement that responsible for the G. The quantitatively constraints would be done in the future work, if we can account for the problems well.

According to these considerations, we did add more detailed discussion of the amount and distribution of melts to cause the seismic velocity drop in the revised manuscript.

Reviewer #2 (Remarks to the Author):

The Gutenberg discontinuity is a seismic velocity discontinuity, often associated with the lithosphere asthenosphere boundary beneath oceanic plates. Its origin and, the physical and chemical properties that define them have remained enigmatic. Presence of partial melt is often invoked as one of the mechanisms to explain the sharp seismic velocity contrast, although the factors affecting melt production beneath older oceans is debated. This paper reports evidence that the melt fractions are enhanced by partial melts associated with recycling of ancient marine sediments and oceanic eclogite/pyroxenites. The authors arrive at this conclusion by studying Mg isotope data of basalts erupted on the northwest Pacific plate (petit spot lavas). The basis of their conclusion that these melts are responsible for the G discontinuity is derived from the observation of a sharp velocity discontinuity by Kawakatsu et al., (2009) on the ~131 Ma year old Pacific plate, close to their study/sampling regions.

Previous seismic studies of the Pacific Lithosphere have argued for the presence of melt closer to the ridges and regions altered by volcanism to explain the G discontinuity (Tharimena et al., 2017; Rychert and Shearer, 2011; Schmerr, 2012). Since mantle melt production is expected to be lower

beneath older lithosphere, a more viable explanation for bulk of the observations of the G discontinuity beneath the Pacific is presence of frozen in melt. In regions of local enrichment such as mantle plumes or subducting slabs, volatiles could stabilize melt accumulation beneath older lithosphere (Sifre et al, 2014). In fact, Tharimena et al. (2017) argued for the presence of carbonatitic melt to explain some of their observations of sharp velocity change beneath older seafloor.

The present study provides further support and evidence to the notion of melt beneath older seafloor and the authors also provide a suitable mechanism to sustain the supply of melt through recycling of oceanic crustal material. This study provides a significant geochemical observation/constraint for seismic discontinuity observations in the older Pacific ocean closer to regions of subduction. As this paper provides important constraints and implications for future observations of the seismic G discontinuity globally, I can recommend this study for publication in Nature Communications provided the authors address these minor comments:

Minor Comments:

1) The base of the plate that translates from a rigid lithosphere to the mobile asthenosphere is called the lithosphere – asthenosphere boundary. Multiple factors influence this boundary including temperature, chemical composition, grain size, anisotropy and even extent of partial melt. While the G discontinuity is the top of the low velocity zone in the upper mantle (Revenaugh and Jordan, 1991). Since the G discontinuity coincides (in some oceanic regions) with the expected LAB depth, it is suggested that G discontinuity is closely related to the LAB.

My bone of contention (although minor) is that the authors use the terms LAB and G discontinuity interchangeably. This may be true for their study as presence of melt would weaken the mantle and therefore imply that the seismic G discontinuity defines the LAB in this region. However, beneath older lithosphere where the discontinuity in seismic velocity is due to say frozen in melt, then the G discontinuity will not coincide with the LAB.

I think the authors should make this distinction in the introduction and conclude the paper with the implication that the G discontinuity in their study region defines the LAB.

Thanks for your suggestions, we have clarify this in the revised manuscript. See line 25 to 28 in the new manuscript.

2) Fig 4: the text labels/annotations and legend are not scaled properly and are a bit hard to read. I

would suggest using similar scaling as in Fig 2.

We have enlarge the labels, annotations and legend in the revised Fig.2.

References:

Kawakatsu et al, 2009; Seismic evidence for sharp lithosphere-asthenosphere boundaries of oceanic plates. Science, 324

Tharimena et al., 2017; Imaging Pacific lithosphere seismic discontinuities – Insights from SS precursor modeling. JGR Solid Earth, 122

Rychert and Shearer, 2011; Imaging the lithosphere – asthenosphere boundary beneath the Pacific using SS waveform modeling., JGR, 116

Schmerr, 2012; The Gutenberg discontinuity: Melt at the lithosphere – asthenosphere boundary, Science, 335

Sifre et al, 2014; Electrical conductivity during incipient melting in the oceanic low-velocity zone, Nature, 509

Revenaugh and Jordan, 1991; Mantle layering from ScS reverberations: 3. The upper mantle. JGR, 96

Reviewer #3 (Remarks to the Author):

Let me start by saying that I am not a high-temperature expert, and was asked to review this primarily for the methods. I make a number comments on these, but this is mainly in the form of tidying up and providing more information.

I have some suggestions for the main manuscript, which largely focus on examining the low-temperature effects more (dolomites and seawater), and also examining diffusion, which to my knowledge has been shown can yield more fractionation than you say.

Thanks for your suggestions. The marine sediments that have low $\delta^{26}\text{Mg}$ are actually caused by the hosted dolomite. We have clarified this in the revised manuscript (See line 110 to 111 in the revised manuscript).

However, we still hold our previous view that at high temperature, the diffusion during the transport of porous flow will not explain our data. Please see the explanation in the following

answers for your comments on Lines 193-200.

Line 61-63: except of course by diffusion processes, which can generate significant fractionation.

See Pogge von Strandmann et al., 2011, 2015; Lai et al., 2015; Hin et al. 2017.

Thanks for your suggestions. Yes, the diffusion processes could lead to larger extend of Mg isotope fractions. However, the reported diffusion-inducing Mg isotopes fractionations in the literatures so far were rightly for the case of metamorphism at low temperature and fluids-activity related (Pogge von Strandmann et al., 2015, ~450°C), or for the interaction between xenocrysts or phenocrysts and hosting magma (Sio et al., 2017; Oeser et al., 2015). These situations are rather different from the processes of mantle partial melting, melt segregation in the mantle, and its fractionation during the ascent. To our knowledge until now there is no case of nature basalt, large amount of bulk rock, were explained as the result of diffusion for their Mg isotope variations.

For Pogge von Strandmann et al. (2011), the author reported the global xenoliths from different background, from craton, off craton to arc. The co-variation of delta Li and deltaMg would most likely reflect the metasomatism by different agency. It would be hard to consider one simple diffusion process to lead to the globally variation. For Pogge von Strandmann et al. (2015), see the explanation in the following part. Lai et al. (2015): While the Li isotopes span large range, the Homoran peridotites show rather limited range of deltaMg. The authors concluded that “The constant d26Mg values of Horoman peridotites suggest that chemical potential gradients caused by melt infiltration were insufficient to drive associated d26Mg fractionation greater than our external precision of 0.03‰”. Finally, the paper by Hin et al. (2017, Nature) reported the fraction of Mg isotopes not by diffusion, but by the accretional vapour loss.

Anyway, we add the reference of Pogge von Strandmann et al. (2015), which should be covered by low-temperature processes. We also clarify the “high temperature magmatic process” inferring to the partial melting and fractionation. See line 63 to 65 in the revised manuscript.

Line 86: it's only really semantics, but the mean d26Mg values have evolved very slightly, with smaller uncertainty, in the past 9 years. See for example Lai et al., 2015 and Hin et al., 2017.

Thanks for your reminding. The range suggested by these two references are actually slightly heavier than the value by Teng et al. (2010). The choice of these values would not change our discussion and conclusions. So we did not demonstrate these mean values in the revised manuscript.

Line 88: “normal” mantle? Primitive mantle?

We are sorry to cause the confusion. We intended to compare the Site B samples with the global MORB and OIB source. We have clarified this in the revised version. See line 95 in the revised manuscript.

Line 96-97: I’m not convinced by this – there almost looks like there is a trend.

Yes, it is not so clear on megascopic, and the R^2 of the linear fitting is relatively low (0.36). But this does not change our conclusion that the alteration is the not reason for the low $\delta^{26}\text{Mg}$. The important thing is that the low $\delta^{26}\text{Mg}$ is preserved in the sample with least alteration. In the revised manuscript, we do not mention a “positive trend”. See line 104 to 106 in the revised manuscript.

Line 103-109: I think this could be explored a bit more, especially in the context of potential dolomites.

Actually, the marine sediments with low $\delta^{26}\text{Mg}$ are mainly due to the precipitation of dolomite (Hu et al., 2017). Such low $\delta^{26}\text{Mg}$ sediments have high radiogenic Pb isotopes (Hu et al., 2017) and low Ce/Pb ratios, thus the assimilation of such sediments could be largely identified by the correlations of Pb isotope and trace element ratios (revised Figure 1).

Yes, in the revised manuscript, we have considered the possible effect of dolomites. Please see line 111 to 114 in the revised manuscript.

Line 112: add ‘g’

We are sorry for this typo error. This has been revised according to your suggestion (see Line 122 in the revised manuscript).

Line 193-200: see Pogge von Strandmann et al 2015 where scenarios are explored where an integrated pore fluid network causes diffusional heterogeneity of over 1%.

Thanks for your suggestions. I know this paper, which reported large Mg isotopic fractionation of the reaction front of an exhumed contact between rocks of subducted crust and serpentinite in the Syros melange zone. The low $\delta^{26}\text{Mg}$ were explained as the results of diffusion through an interconnected pore fluid network at $\sim 430^\circ\text{C}$. The authors suggest that “Grain boundary network diffusion would create an isotopically heterogeneous fluid that has light isotope ratios at the boundary of the minerals (dominantly chlorite) growing by diffusive supply of Mg”. The critical point for this model is that the authors did not consider the re-equilibrium between the

parous fluids and the matrix body, which would buffer the Mg isotopic fractionation caused by diffusion, because at the such low temperature the diffusion in the solid is too sluggish. This solution would be reasonable for this case.

However, it could not be applied to our case because of the following reasons. The high ambient mantle temperature ($\sim 1350^{\circ}\text{C}$) would result in: 1) the fast diffusion within the minerals, which would reset the Mg isotopic fractionation of the parous flow caused by diffusion during their transport; 2) extensive the reaction of the melt with the mantle rock during the melt (initially it is carbonatite in our case) transport, which transfer the carbonatite to carbonated silicates. Due to the two processes above, the Mg isotopic composition of the parous flow in our case would be buffered by the ambient mantle largely. This would be the partial reason why globally the $\delta^{26}\text{Mg}$ of both MORB and OIB are consistent with the mantle peridotite (except the source heterogeneity for few cases, like Pitcairn EM1 basalts, Wang et al., 2019).

Thus, we still hold the previous view that during the melt percolation within the asthenosphere, no significant Mg isotopic fractionation could be caused by diffusion. To the contrary, we suggest it is the buffering by the ambient mantle, the low $\delta^{26}\text{Mg}$ of the carbonatite melt inherited from the mantle source was erased. We have added the brief explanations above to the revised manuscript (see lines from 210 to 215).

Methods: given that this is an NComms paper, I think there needs to be more description of the method. So, for example, give numbers for the column yields. The $\delta^{26}\text{Mg}$ values for the USGS standards should definitely be given, as should be their reproducibility. At the moment you only provide information for BHVO-2 and BCR-2, and not any of the other standards you mention. This should also include the 'in-house' standards, because, at least in the case of CAM-1 and SRM980, are not actual 'in-house' standards, but have been measured by a large number of groups.

I would also like to know exactly what the N means in your Supp. Table 4: is that for complete reanalyses (i.e. including dissolution and chemistry), or analysis repeats during a single run, or what?

The column yields are always better than 99.5%.

We are sorry in the previous version we made a mistake on the description of the Mg isotope method. The approach reported by An et al. (2014) conducted many analyses on international standards, that is in order to build the standard procedure and compare with other labs. For our samples, at 2017, the method in this lab has been routine, so we did not actually use so many standards. In our analysis session, we used the BHVO-2 and BCR-2 as the standards to see whether our dissolution, column chemistry and MS analysis could produce results comparable with other international labs (accuracy), and used in-house standard (IGGMg1, IEE, they are synthetic standard) bracketed the unknown samples to see the repeatability.

We have revised the description about this point in the revised Method (see line 259 to 260). We also added the repeated analysis on IGGMg1 and IEE in the revised Supplementary Table 4. In this table, n represent the times of repeat analysis by MS for the same Mg purified cuts, which is mentioned in the note of table. Actually, we also have marked the complete re-analyses including dissolution and chemistry by “replicate” in this table.

I would also suggest that you expand on the ‘recommended’ values for BHVO-2 and BCR-2. The number of An et al are quite heavy compared to many other studies, especially in light of the precision you are reporting. You could look at the compilation in Pogge von Strandmann et al., 2011, or indeed a swift perusal of GeoRem shows that there are 38 different publications for $\delta^{26}\text{Mg}$ for BCR-2 and 31 for BHVO-2.

Thanks for your suggestions. We have expanded the recommended values for these two standards in the revised Supplementary Table 4.

We checked the recent compilation of analysis on Mg isotope standards by Teng (2017, RMG, Table 1). We see for the BHVO-2, the analyzed value by Pogge von Strandmann et al. (2008, 2011) is $-0.25 \pm 0.11\text{‰}$, and $-0.26 \pm 0.06\text{‰}$, respectively, while the recommended value by Teng is $-0.24 \pm 0.08\text{‰}$. These values are comparable with An’s recommended value for this standard ($-0.22 \pm 0.04\text{‰}$). For BCR-2, the analyzed values from different labs vary from $-0.16 \pm 0.01\text{‰}$ to $-0.33 \pm 0.04\text{‰}$. Yes, An’s value is heavier than the high limit value, but is still consistent with $-0.16 \pm 0.11\text{‰}$ (Bourdon et al., 2010), -0.19 ± 0.02 (Baker et al., 2005), $-0.17 \pm 0.35\text{‰}$ (Bizzarro et al., 2004), $-0.14 \pm 0.11\text{‰}$ (Wombacher et al., 2009), -0.16 ± 0.11 (Tipper et al., 2008), $-0.25 \pm 0.06\text{‰}$ (Pogge von Strandmann et al., 2011), and $-0.20 \pm 0.07\text{‰}$ (Wimpenny et al., 2014b). As suggested by An et al. (2014), the discrepancy with the lighter values (< -0.3) would be possibly

due to the “dry” plasma was used for the latter. Anyway, during our session, the analyzed BRC-2 value is $-0.22 \pm 0.06\%$, which is consistent with many published values within errors.

Lastly, you don't seem to have repeated any of the standards (?), so it is difficult to provide a true test of combined accuracy and precision.

Actually, during our analysis session, the international standards were analyzed and compared with the recommended value. If the measured value for these standards (BCR-2, BHVO-2) are consistent with the recommended value within the uncertainty (2sd of multiple analysis for the same purified cuts), we say our results should be “accuracy”, and comparable crossing the labs. The precision could be seen by the repeated analysis of the in-house standards and selected samples (repeat here means complete re-do of dissolution, column chemistry, and analysis by MS).

We have added the repeated analysis of in house standards in the Supplementary Table 4.

REVIEWERS' COMMENTS:

Reviewer #1 (Remarks to the Author):

I think the authors have done a very good job of addressing all reviews and the changes to the manuscript are fine. I would be happy to see it go forward from here.

Reviewer #2 (Remarks to the Author):

The authors addressed concerns raised in the previous review. This is an interesting paper and provides important constraints and implications for future observations of the seismic G discontinuity globally. I recommend this study for publication in Nature Communications.

Reviewer #4 (Remarks to the Author):

This paper reports new Mg isotope data for 15 lavas from three seamounts located in the Western Pacific ocean, close to an area where a so-called Gutenberg discontinuity has been seismically observed. The authors find that many of these samples extend to Mg isotope compositions lighter than the average mantle (as determined by global MORB and OIB samples). They argue that the light values originate from the mantle, rather than from crustal processing. In addition, the authors use cross-plots with Pb isotopes and trace element ratios to develop an interpretation whereby the light Mg isotopes are derived from carbonated eclogite (which was subsequently de-carbonated) whereas the mantle-like Mg isotopes derive from a source that was contaminated by sediments (not clear to me if these were also carbonated or not). With this interpretation the authors infer a linkage between the Gutenberg discontinuity and carbonated melts at the base of the lithosphere-asthenosphere boundary (at least in the Western Pacific).

In general, I like the idea behind this paper: namely that Mg isotopes can potentially be used to trace carbonate contamination in the mantle because of the very light Mg isotope compositions found in many carbonates compared with the very homogenous Mg isotopes found in almost all mantle derived magmas. However, there is one major issue with the concept and that is the mass balance of carbonate addition to the mantle has some well-known collateral effects which the authors here (and others before them) tend to pay little attention to. For example, Sr concentrations are very high and Sr isotope compositions very radiogenic in essentially all surface carbonates and one would expect substantially elevated Sr/Nd (relative to mantle) and very radiogenic Sr isotopes in mantle reservoirs contaminated by carbonates. The authors do not even discuss this issue in the paper. Furthermore, the amount of carbonate required to produce significant Mg isotope anomalies from mixing is very substantial because of the very high Mg concentrations in the mantle. This would leave the contaminated region with very high carbonate concentrations. Here the authors seem to get around this problem by inferring that the isotopically light eclogite was de-carbonated after having the Mg isotopes overprinted by carbonation. I must admit that I find this explanation somewhat ad-hoc and certainly (as also pointed out by several previous reviewers) quite complicated. In the response letter the authors underline that they think there are significant correlations between Mg isotopes, Pb isotopes, and trace element ratios, but there are no quantification of such mixing lines in figure 4 (ie actual mixing lines between different endmembers), merely the authors indicate some dashed lines to lead the eye towards mixing lines that several reviewers question the validity of. I would tend to agree with the previous reviewers that the correlations with Mg isotopes are pretty weak often defined by one outlier, the removal of which would eliminate the correlations.

On a related note, I am also concerned about how quick the authors are to dismiss the possibility that alteration could have modified the Mg isotopes. They simply state that the least altered sample (although they present no evidence to support that statement) has the lightest Mg isotopes and that there is no correlation with LOI. However, I find neither of these arguments convincing

(like several reviewers before me) and there are actually ways in which one can use the trace element data the authors have already generated to further interrogate the possibility of alteration. For example, Th/U and Ba/Rb ratios are typically quite constant in mantle derived lavas, whereas both U and Rb are added during low-T hydrothermal alteration. It is notable that several of the samples with the lightest Mg isotope compositions recorded in this study also have low Th/U and Ba/Rb relative to the remainder of the samples. It is particularly relevant that the samples with light Mg isotopes have low Th/U compared with average mantle because that is the opposite to what should be found for de-hydrated/de-carbonated eclogite where U is more mobile than Th. In other words, an eclogite reservoir, like the one inferred by the authors, should have high Th/U, not the low values observed. This does not prove that alteration modified the Mg isotopes, but it certainly supports such an interpretation. Submarine basalts that have been submerged in seawater for millions of years are notoriously tricky to be 100% sure of as entirely pristine. At least for elements that are easily modified by alteration such as the alkali metals, U and to some extent Mg.

I am not sure where this leaves the authors. On the one hand, I would tend to agree that at least some of the light values are probably reflecting the Mg isotope composition of the mantle source, but its just hard to tell beyond any doubt. I would suggest the authors at the very least have a much expanded discussion of the alteration possibility with a few figures to go with it. That way readers can at least make up their minds by evaluating all the evidence at hand.

A few additional things to address:

Line 125: It says Mg isotope difference here. What direction of isotope fractionation? Is it even it the right direction to explain the data?

Lines 132-136: I dont understand this discussion of ilmenite and its effect on Mg isotopes. Why would ilmenite matter when that mineral contains virtually no Mg? Also, I dont think there is any logic to comparing terrestrial basalts with lunar basalts here given how different these two mantles are.

Line 158 (and figure 2c-2d): these mixing lines must be wrong. How can you mix carbonatite into a mantle source and get two different Pb isotope compositions for the same amount of carbonatite? This is what the dashed lines in figures 2c and 2d show. Also, mixing lines should be smooth, not jagged like that, which is due to the authors just connecting incremental points rather than the full range of mixing proportions.

Line 160: The carbonate content of the primitive melt is an important piece of evidence that I think needs to be elaborated upon. What is that based on?

Line 182: You need some references for this statement. Also, why does it necessarily have to get carbonated? There are clearly mass transfer processes for subducted slabs that could be associated with Mg isotope fractionation leaving a light residue that have nothing to do with carbonation. There are some arcs that have revealed heavy Mg isotopes and I do not believe these results have had anything to do with carbonates or carbonation. This issue also goes back to the statement of 10% carbonate in the parental melts. How well constrained is that? And is it really required that these light Mg isotope values (if indeed they are mantle derived) have anything to do with carbonate?

Line 220 (end of line): something is clearly missing here? Some text?

REVIEWERS' COMMENTS:

Reviewer #1 (Remarks to the Author):

I think the authors have done a very good job of addressing all reviews and the changes to the manuscript are fine. I would be happy to see it go forward from here.

Reviewer #2 (Remarks to the Author):

The authors addressed concerns raised in the previous review. This is an interesting paper and provides important constraints and implications for future observations of the seismic G discontinuity globally. I recommend this study for publication in Nature Communications.

Reviewer #4 (Remarks to the Author):

This paper reports new Mg isotope data for 15 lavas from three seamounts located in the Western Pacific ocean, close to an area where a so-called Gutenberg discontinuity has been seismically observed. The authors find that many of these samples extend to Mg isotope compositions lighter than the average mantle (as determined by global MORB and OIB samples). They argue that the light values originate from the mantle, rather than from crustal processing. In addition, the authors use cross-plots with Pb isotopes and trace element ratios to develop an interpretation whereby the light Mg isotopes are derived from carbonated eclogite (which was subsequently de-carbonated) whereas the mantle-like Mg isotopes derive from a source that was contaminated by sediments (not clear to me if these were also carbonated or not). With this interpretation the authors infer a linkage between the Gutenberg discontinuity and carbonated melts at the base of the lithosphere-asthenosphere boundary (at least in the Western Pacific).

In general, I like the idea behind this paper: namely that Mg isotopes can potentially be used to trace carbonate contamination in the mantle because of the very light Mg isotope compositions found in many carbonates compared with the very homogenous Mg isotopes found in almost all mantle derived magmas. However, there is one major issue with the concept and that is the mass balance of carbonate addition to the mantle has some well-known collateral effects which the authors here (and others before them) tend to pay little attention to. For example, Sr concentrations are very high and Sr isotope compositions very radiogenic in essentially all surface carbonates and one would expect substantially elevated Sr/Nd (relative to mantle) and very radiogenic Sr isotopes in mantle reservoirs contaminated by carbonates. The authors do not even discuss this issue in the paper.

Thanks for your comments. Yes, you are right, but this situation is mainly applicable for the carbonate consisting of calcite. However, the calcite has very low MgO, which would be not easy to change the Mg isotopic composition of the mantle rock. Actually, in many literatures (Huang and Xiao et al., 2016; Li et al., 2017), the most possible candidate to change the Mg isotopes of the mantle significantly but not to change the Sr isotopes, is magnesite, which has high MgO

content up to 47.6wt.% and Sr less than 2 ppm (Huang and Xiao, 2016 IGR). Although the magnesite would not be the main specie of the carbonate in the oceanic crust in shallow subduction zone, the dolomite or calcite would react with the silicate mineral and transformed to magnesite when the slab subducted into deep mantle (Huang and Xiao, 2016 IGR).

Furthermore, the amount of carbonate required to produce significant Mg isotope anomalies from mixing is very substantial because of the very high Mg concentrations in the mantle. This would leave the contaminated region with very high carbonate concentrations. Here the authors seem to get around this problem by inferring that the isotopically light eclogite was de-carbonated after having the Mg isotopes overprinted by carbonation. I must admit that I find this explanation somewhat ad-hoc and certainly (as also pointed out by several previous reviewers) quite complicated.

The reason why we infer that the light Mg isotopic endmember samples were not produced by a carbonated mantle source (e.g. it means that considerable amount of CO₂ must be present when the mantle melted) is, this end member (in text, it infers to high-²⁰⁶Pb/²⁰⁴Pb end-member) does not show high Zr/Hf, Nb/Ta ratios, and low Ti/Eu ratio, which are the indicative characteristics of the carbonated partial melts (carbonatite and carbonated silicates). A more suitable candidate, who holds the feature of low delta²⁶Mg but no CO₂, is the eclogites/pyroxenite. It has been widely reported that this material can meet those requirements (Wang SJ et al., 2015, *Geology*; Wang XJ et al., 2018, *PNAS*). Other than the de-carbonated after having the Mg isotopes overprinted by carbonation, the eclogites/pyroxenite could get light Mg isotopes through Fe-Mg exchange with the hosting peridotite. In addition, the eclogites/pyroxenite is easy to make the partial melts with high MgO, because some of them can have considerably high MgO.

We have expanded this explanation in the revised manuscript (Line 177 to 180).

In the response letter the authors underline that they think there are significant correlations between Mg isotopes, Pb isotopes, and trace element ratios, but there are no quantification of such mixing lines in figure 4 (ie actual mixing lines between different endmembers), merely the authors indicate some dashed lines to lead the eye towards mixing lines that several reviewers question the validity of. I would tend to agree with the previous reviewers that the correlations with Mg isotopes are pretty weak often defined by one outlier, the removal of which would eliminate the correlations.

Actually, the previous reviewers did not question the validity of our mixing models, they just concern what kind of carbonatite was used in the calculation. In the previous revision, we have explained this. Actually, the mixing modeling we did is to argue against the possibility that these magmas were produced by partial melting of a series of mantle sources formed by a common source metasomatized by one carbonatite (this could also be known as “source contamination”).

I totally understand if we can incorporate more quantitative constraints on the mantle source constitutes and the specific partial melting partial melting, this work would be largely improved. However, as we replied to the other reviewers, we can't not do any meaningful quantification on those topics at this stage, because many parameters used in the quantification could not be constrained. For instance, we can't know the specific composition of the pyroxenite/eclogites (e.g. MgO content, Na₂O+K₂O content), which would largely control the solidus temperature. About the quantification on the magma mixing, in sure it can be done, but don't have much helps. The

more important thing is there is two end members.

Yes, the correlations of Mg isotopes with U/Pb, Sm/Yb ratios may be controlled by one point. But, if you look into Figure 4a-b, except one sample #1390R10 (which is solely outside in line in Figure 3b), all the samples from three sites show similar trends.

On a related note, I am also concerned about how quick the authors are to dismiss the possibility that alteration could have modified the Mg isotopes. They simply state that the least altered sample (although they present no evidence to support that statement) has the lightest Mg isotopes and that there is no correlation with LOI. However, I find neither of these arguments convincing (like several reviewers before me) and there are actually ways in which one can use the trace element data the authors have already generated to further interrogate the possibility of alteration. For example, Th/U and Ba/Rb ratios are typically quite constant in mantle derived lavas, whereas both U and Rb are added during low-T hydrothermal alteration. It is notable that several of the samples with the lightest Mg isotope compositions recorded in this study also have low Th/U and Ba/Rb relative to the remainder of the samples. It is particularly relevant that the samples with light Mg isotopes have low Th/U compared with average mantle because that is the opposite to what should be found for de-hydrated/de-carbonated eclogite where U is more mobile than Th. In other words, an eclogite reservoir, like the one inferred by the authors, should have high Th/U, not the low values observed. This does not prove that alteration modified the Mg isotopes, but it certainly supports such an interpretation. Submarine basalts that have been submerged in seawater for millions of years are notoriously tricky to be 100% sure of as entirely pristine. At least for elements that are easily modified by alteration such as the alkali metals, U and to some extent Mg. I am not sure where this leaves the authors. On the one hand, I would tend to agree that at least some of the light values are probably reflecting the Mg isotope composition of the mantle source, but its just hard to tell beyond any doubt. I would suggest the authors at the very least have a much expanded discussion of the alteration possibility with a few figures to go with it. That way readers can at least make up their minds by evaluating all the evidence at hand.

Thanks very much for such thoroughly comments on our discussion on the role of the alteration. We have check the Th/U and Ba/Rb ratios and their correlations with LOI, for samples from Site A (simply because this site has the largest amount of samples). As shown in the following Figure R1, these two ratios are both positively correlated with LOI, with $R^2 > 0.7$. These trends are rightly opposite to that you suggested if the alteration did large effects, which strongly argue for that the low Th/U ratios for the samples with light $\delta^{26}\text{Mg}$ were their own characteristics. In addition, as shown in the $\delta^{26}\text{Mg}$ vs. Ba/Rb plotting, the low Ba/Rb samples (~20) show rather large range of $\delta^{26}\text{Mg}$ (Figure R1c). We put this line in the main text (line 110-113).

We also check the previous work on the Mg isotopic composition of the altered oceanic crust (samples from the IODP drill, Huang et al., 2015), which shows that the oceanic crust with LOI up to 4.7wt.% still hold the mantle-like value (see Figure R2). Actually, the pure water-rock interaction does not change the Mg isotopes of mafic rocks in significance, because the mafic rocks have much higher MgO content than the seawater. As shown by Huang et al. (2015) and the recent investigation on marine sediments (Hu Yan et al., 2018, CG), the low $\delta^{26}\text{Mg}$ signature during the low temperature process can be formed only by precipitation of carbonate (calcite or

dolomite). During our sample preparation, we have cut off the surface samples, crash the internal part into small piece, and selected very carefully. Note that, if there is considerable amount of such carbonate in our selected materials, the LOI must be high!

Combining all these considerations, we keep our previous suggestion, the alteration does not change the Mg isotopes.

Figure R1. LOI vs. Ba/Rb, Th/U, and their comparisons with delta26Mg for samples from site A. (We also put this figure in the Supplementary Fig. 3).

Figure R2. LOI vs. delta26Mg of altered oceanic crust (IODP site 1256), figure from Huang et al. (2015). *Lithos*, 231 (2015) 53–61. The grey region show the Mg isotopes range of the mantle.

A few additional things to address:

Line 125: It says Mg isotope difference here. What direction of isotope fractionation? Is it even in the right direction to explain the data?

Sorry for the nonstandard usage. We have substituted “difference” with “fractionation”, and add “+” before 0.1‰, which would clearly demonstrate the fractionation direction. See line 127 to 128.

Lines 132-136: I don't understand this discussion of ilmenite and its effect on Mg isotopes. Why would ilmenite matter when that mineral contains virtually no Mg? Also, I don't think there is any

logic to comparing terrestrial basalts with lunar basalts here given how different these two mantles are.

This is mainly because we observe a negative correlation between $\delta^{26}\text{Mg}$ and TiO_2 , which means that the light Mg seems tend to high TiO_2 (Figure 3b). Yes, the ilmenites have low Mg content, but if they are rich in the source, the partial melting would be also reset for their Mg isotopes. We agree with you that this possibility for earth mantle is rather low, but in order to make a discussion as comprehensive as possible, it's better to keep this part of discussion there.

Line 158 (and figure 2c-2d): these mixing lines must be wrong. How can you mix carbonatite into a mantle source and get two different Pb isotope compositions for the same amount of carbonatite? This is what the dashed lines in figures 2c and 2d show. Also, mixing lines should be smooth, not jagged like that, which is due to the authors just connecting incremental points rather than the full range of mixing proportions.

We are so sorry for such a stupid mistake. We have checked the modeling, the mixing lines are right, but the number markers are wrong. We have fixed this in the version. We also added some points to make the line smoother.

Line 160: The carbonate content of the primitive melt is an important piece of evidence that I think needs to be elaborated upon. What is that based on?

It is the content of CO_2 in the primary magma. It is inferred from the direct measurement of CO_2 content in the quench glass and the modeling the degassing process based on the $\text{CO}_2\text{-H}_2\text{O}$ solubility. We have added this explanation to the main text (see line 163-165).

Line 182: You need some references for this statement. Also, why does it necessarily have to get carbonated? There are clearly mass transfer processes for subducted slabs that could be associated with Mg isotope fractionation leaving a light residue that have nothing to do with carbonation. There are some arcs that have revealed heavy Mg isotopes and I do not believe these results have had anything to do with carbonates or carbonation. This issue also goes back to the statement of 10% carbonate in the parental melts. How well constrained is that? And is it really required that these light Mg isotope values (if indeed they are mantle derived) have anything to do with carbonate?

We also add the reference to the statement of high Ce/Pb and U/Pb ratios for the eclogites (see line 188).

We agree with you that the low $\delta^{26}\text{Mg}$ of eclogite/pyroxenite does not necessarily directed correlated with the carbonate. It has been revealed that the Mg exchange between the eclogite/pyroxenite and the hosting mantle during the subduction could also lead to low $\delta^{26}\text{Mg}$ for the eclogite/pyroxenite (Wang et al., 2015). We have added this possibility to the revised discussion (see lines 177-180). Addition of this eclogite candidate does not change our discussion and conclusion. The main point of our discussion is that, the high $^{206}\text{Pb}/^{204}\text{Pb}$ and low $\delta^{26}\text{Mg}$ endmember does not fit the trace elemental features for a carbonated partial melting, which would lead to rather high Zr/Hf and Nb/Ta ratios.

About the 10% statement. This value is the CO_2 content by weight in the primary magma, which was inferred by the measured CO_2 and H_2O content of the quench glass and the $\text{CO}_2\text{-H}_2\text{O}$ solubility model. We just mention it as it is one of the clues that some magmas are carbonated

(silicate).

Line 220 (end of line): something is clearly missing here? Some text?

We are sorry for this typewriting mistake, which would occur at the revision mode of the Word.

We have fixed this in the new manuscript (line 222-225).